# Semaphorin 3f and post-embryonic regulation of retinal progenitors

**Amira Kalifa, Carrie L. Hehr, Katelyn L. Shewchuk®,
Risa Mori-Kreiner, Shaelene Standing, Rami Halabi®, Sarah McFarlane®***

Department of Cell Biology and Anatomy, Hotchkiss Brain Institute, Alberta Children's Hospital Research Institute, University of Calgary, Calgary, Alberta, Canada

* smcfarla@ucalgary.ca

## Abstract

Neural progenitors produce specific cell types that form the circuits of the nervous system. Extrinsic signals regulate both progenitor proliferation and the production of specific neuron types. Where progenitors reside within a progenitor niche determines to which of these signals they are exposed, and thus likely has important consequences on the progeny they produce. Little is known, however, of the signals that govern progenitor location within the niche. Here we show that a member of the Class III family of secreted Semaphorins, Semaphorin3fa (Sema3fa), is required for the orderly arrangement of progenitors with a niche present in the periphery of the larval and adult retina of zebrafish, the ciliary marginal zone (CMZ). CMZ progenitors express mRNAs for various Sema3 receptors, including for *nrp2a*, *nrp2b* and *plxna1*. Loss of Sema3fa in a previously characterized CRISPR/Cas9-generated *sema3fa* mutant allele (*sema3fa^ca304^*) results in a reduced eye size in juvenile fish, implicating Sema3fa in the ongoing production of retinal cells by the CMZ. Larval progenitors show altered cell cycle parameters, and the spatial organization of functionally distinct progenitors is disrupted, as is the generation of retinal cell types in the appropriate proportions and numbers. Our data support a model whereby Sema3fa secreted by CMZ progenitors reduces adhesive interactions to allow for smooth progression of progenitors through the niche, ensuring progenitors receive the correct recipe of extrinsic signals to secure the proper generation of new retinal circuits.

## Author summary

After embryonic development is complete, neural stem cells and progenitors remain in tissue domains called stem cell niches within select regions of the nervous system. The progenitors in these niches produce new neurons for the ongoing growth and remodelling of existing neural circuits. A niche is present in the retina of non-mammalian vertebrates, such as zebrafish, called the ciliary

**Data availability statement:** Data in graphs and within the text have been deposited in a Dryad repository: DOI: https://doi.org/10.5061/dryad.pc866t21r.

**Funding:** The work was supported by a Canadian Institutes for Health Research project grant #PJT-178044 to S.M. Support (no numbered grants) for students included an Alberta Children's Hospital Research Institute graduate studentship for A.K., a Hotchkiss Brain Institute Recruitment Award and a Spinal Cord Nerve Injury scholarship to K.LS, a Natural Sciences and Engineering Research Council Undergraduate Student Research Award to S.S, and Hotchkiss Brain Institute T. Chen Fong and Alberta Innovates studentships to R.H. The funders had no role in study design, data collection and analysis, decision to publish, or preparation of the manuscript.

**Competing interests:** The authors have declared that no competing interests exist.

marginal zone (CMZ). The CMZ sits in the retinal periphery, encircling the lens, and produces new neurons to allow ongoing formation of retinal circuits as the eye continues to grow over the life span. The progenitors are tightly packed within the niche and organized in a spatial fashion based on both the genes they express, and their maturity and function. Our data suggests that in zebrafish this organization depends on a protein called Semaphorin3fa that the progenitors both secrete and respond to. Without Semaphorin3fa, the organization breaks down, and consequently progenitors make fewer and the wrong types of neurons. Our data support a model where Semaphorin3fa reduces how tightly progenitors stick to one another as they mature. This feature allows maturing progenitors to readily move through the CMZ so that progenitors receive appropriate instructions about which progeny to produce. We propose our work identifies for the first time a niche-autonomous molecular mechanism for controlling the organization of neural progenitors and their ability to produce neurons in the right numbers and types.

## Introduction

Where a cell sits within a progenitor niche exposes it to a unique cadre of extrinsic signals, which guide proliferative and neurogenic behaviour. The involvement of key secreted factors, such as Fibroblast Growth Factors, Sonic Hedgehog (Shh), and Wnts in progenitor cell fate choice is known [1]. Poorly understood, however, are the molecules that determine the location of progenitors within a proliferative niche, despite the deciding role location likely plays in producing the appropriate types and numbers of cells.

The vertebrate retina is an evolutionary conserved, organized structure of various cell types tasked to convert visual stimuli into electrical impulses [2]. In the embryo, the genesis and differentiation of vertebrate retinal neurons occurs in organized temporal waves regulated by both intrinsic and extrinsic signals [3]. An interesting model to understand how spatial information regulates progenitor behaviour is the post-embryonic retina of non-mammalian vertebrates, which contains a spatially organized progenitor niche, the ciliary marginal zone (CMZ). CMZ progenitors provide cells for new circuits at the retinal periphery, causing lifelong growth of the eye [4,5]. The CMZ is organized into domains of progenitors in different functional states. Moving from the periphery centrally towards the retina there are stem cells, proliferative progenitors, fate-restricted progenitors, and newly born neurons [6]. This organization presumably ensures appropriate progenitor proliferation and neurogenesis. The nature of the intrinsic and extrinsic signals that establish and maintain the organization, however, are unknown.

Semaphorins (Semas) are extracellular signaling molecules that guide migrating neurons, axons and endothelial cells [7]. Class III Semas (Sema3s) are secreted molecules that act via Neuropilin (Nrp) and Plexin (Plxn) receptors [8]. Sema3s are best known as guidance molecules, but they also regulate neural progenitor behaviour.

For instance, Sema3d knockdown causes cell cycle arrest of zebrafish neural crest progenitors [9], SEMA3B orients the mitotic spindle of mouse spinal cord progenitors [10], while SEMA3F in the cerebrospinal fluid regulates mouse cortical neurogenesis [11].

Here, we investigate a role for Sema3fa in CMZ progenitor behaviour by taking advantage of a zebrafish *sema3fa* null allele, *sema3fa*[ca304/ca304] [12]. We find CMZ progenitors express mRNA for *sema3fa* and its receptors, *nrp2a/b, plxna1a/b* and *plxna2* [13,14]. The CMZ is not impacted grossly by the loss of Sema3fa, but the compact domains of proliferative and zonal markers of the wild type (WT) CMZ are dispersed in mutants. Radial organized cohorts of cells produced by the mutant CMZ are smaller in size and more variable with respect their cellular makeup, features that may explain why mutant eyes grow less over the larval period. These data, together with the finding that adhesive mechanisms are upregulated in the mutant CMZ, leads us to a model whereby Sema3fa promotes smooth central-wards movement of progenitors through the CMZ niche by limiting adhesion between progenitors.

## Results

### The CMZ expresses mRNA for *sema3fa* and its known receptors

We reported previously that *sema3fa* mRNA is present within the zebrafish CMZ [12], a stem cell niche in the peripheral retina that allows for ongoing growth of the zebrafish eye through adulthood [15]. Here, we used whole mount (WM) and fluorescent slide *in situ* hybridization (ISH) to further characterize *sema3fa* mRNA in the CMZ, and to identify potential receptors through which the protein acts. *sema3fa* mRNA is present in CMZ progenitors as this domain emerges at about 48 hours post-fertilization (hpf) [16], and remained through the end of embryogenesis (72 hpf) (Fig 1A) and on to larval stages (Fig 1B and 1C). *sema3fa* mRNA was present throughout much of the CMZ, though appeared to be absent from the most distal CMZ where the stem cells reside (Fig 1A–1C, black arrows). Indeed, double fluorescent ISH (FISH) for *cyclin d1* (*ccnd1*), a marker of proliferative progenitors in the CMZ [17], alongside *sema3fa* revealed considerable overlap, with little or no expression in the distal retinal stem cell domain (S1A and S1B Fig; white arrows). Of note, while we reported previously that *sema3fb* is expressed along with *sema3fa* within the temporal zebrafish eye primordium at 18 hpf, expression of *sema3fb* is lost subsequently in the embryonic neural retina and mRNA is not present in the post-embryonic CMZ [18].

Sema3s signal by binding to a Nrp-PlxnA heteroreceptor complex [14,19]. To determine if progenitors within the CMZ have the necessary receptors to respond to Sema3fa, we asked by WM ISH whether mRNAs for Nrp and Plxn were expressed by CMZ progenitors at 72 hpf and 7 days post fertilization (dpf). We focused on receptors that show high binding affinity to Sema3fa and are thought to mediate its actions [13,14]; Nrp2 and Plxna. We examined mRNA expression of *nrp2* (*nrp2a,* and *nrp2b*) and *plxna* (*plxna1a, plxna1b, plxna2, plxna3 and plxna4*), and then performed plastic tissue sectioning to investigate expression within the CMZ.

At 48 hpf, *nrp2a* and *nrp2b* mRNAs are expressed by cells of the retinal ganglion cell (RGC) layer, and in the case of *nrp2b* by the CMZ [20]. The mRNAs of the receptors were generally restricted within the CMZ. In transverse sections, *nrp2a* (Fig 1D–1F) and *nrp2b* (Fig 1G–1I and 1P) were expressed in the central CMZ at both 72 hpf (Fig 1D, 1G and 1P) and 7 dpf (Fig 1E, 1F, 1H and 1I), and were reduced or absent from the distal CMZ (Fig 1F and 1I; arrows). Of note, paralogs *nrp1a* and *nrp1b* were also expressed by CMZ progenitors at both 72 hpf and 7 dpf (S1C–S1F Fig).

Zebrafish have two paralogs of *plxnal, plxna1a* and *plxna1b.* In the early optic vesicle (18 hpf), *plxna1a* is expressed by retinal pigment epithelial (RPE) and dorsal eye progenitors and *plxna1b* by some neural progenitors [21]. In transverse sections, mRNAs for the two paralogs were present throughout the CMZ at both 72 hpf and 7 dpf (Fig 1J–1O). Of note, we found mRNA for *plxna2* (S1G Fig) and *plxna3* [14,22], but not *plxna4* (S1I Fig), within the CMZ at 72 hpf, while at 7 dpf mRNAs for all three receptors were present in the CMZ (S1H and S1J Fig). Thus, CMZ progenitors express the necessary receptors to respond to the Sema3fa the CMZ cells produce. To determine whether cycling progenitors might respond to Sema3fa, we performed double FISH for *ccnd1* and *nrp2b*, the latter as a representative receptor that

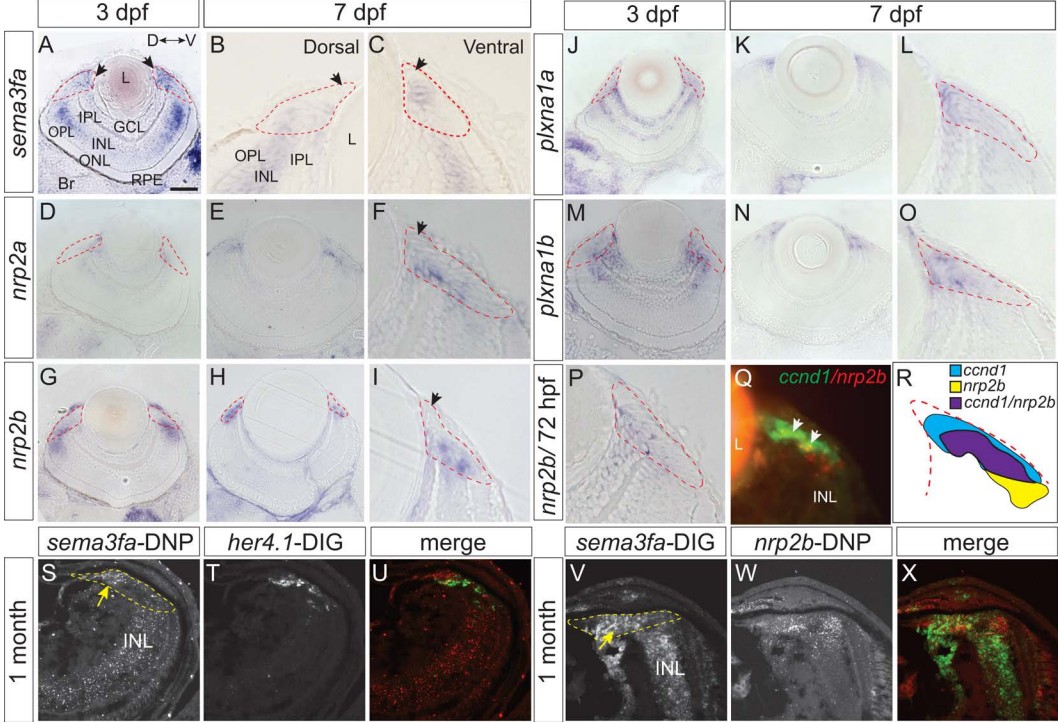

**Fig 1. Expression of mRNAs for Sema3fa and its receptors in the larval and juvenile CMZ.** Plastic sections through the eyes of WM ISH of 3 (A, D, G, J, M, P) and 7 (B, C, E, F, H, I, K, L, N, O) dpf larvae for *sema3fa* (A-C) and potential receptors, *nrp2a* **(D-F)**, *nrp2b* **(G-I, P)**, *plxna1a* **(J-L)**, and *plxna1b* **(M-O)**. The CMZ (red dashed lines) is shown enlarged at 7 dpf (B, C, F, I, L, O). Peripheral-most CMZ lacks expression (arrows B, C, F, I). **Q)** Double FISH for *nrp2b* (red) and *ccnd1* (green) mRNA in the 72 hpf CMZ. **R)** Schematic representing general organization of *ccnd1* (blue) and *nrp2b* (yellow) expression in the CMZ. The two expression domains exhibit some overlap (purple). **S-X)** Double FISH on sections of 1-month retinas with a DNP riboprobe for *sema3fa* (S) and DIG probe for *her4.1* (T; merge in U), and with a DIG probe for *sema3fa* (V) and DNP probe for *nrp2b* (W; merge in X). Expression of *sema3fa* in CMZ (dotted yellow line) indicated by yellow arrows. Orientation bar in A applies to A, D, E, G, H, J, K, M, N. D, dorsal; GCL, ganglion cell layer; INL, inner nuclear layer; IPL, inner plexiform layer; L, lens; ONL, outer nuclear layer; outer plexiform layer, OPL; RPE, retinal pigment epithelium; V, ventral. Scale bar in A is 50 μm for A, D, E, G, H, J, K, M, N and 10 μm for all other panels.

transduces Sema3fa signals both *in vitro* and *in vivo* [11,14,23]. *nrp2b* FISH signal partially overlapped with that of *ccnd1*; *nrp2b* mRNA was expressed by cells on the central side of the *ccnd1* mRNA domain, and by more centrally located CMZ cells that presumably were exiting or had exited the cell cycle (Fig 1Q and 1R). These data argue that mRNAs for the receptors are generally in cells closer to the differentiated central retina than those that express *sema3fa* mRNA (Fig 1R). Importantly, in one-month old juvenile fish, *sema3fa* continued to be expressed in the proliferative CMZ marked by *her4.1* (Fig 1S–1U), alongside progenitors that express the mRNA for the Nrp2b receptor that would allow them to respond to Sema3fa (Fig 1V–1X).

## Eye size is reduced in *sema3fa*ca304 fish

To assess a role for Sema3fa within the CMZ we took advantage of a CRISPR mutant allele we generated and previously characterized, *sema3fa*$^{ca304/ca304}$, which has a 2 base pair deletion in exon 2 that introduces a frame shift and is predicted to produce a truncated protein; Western analysis indicates that the mutant is null for Sema3fa protein [12]. Recently, we found that loss of Sema3fa results in a mild embryonic eye phenotype, with a small but significant disruption in the production of amacrine cells specific to the dorso-temporal retina [18]. Of note, the retinas of the mutants are properly laminated by 72 hpf and the RGC layer, and outer inner nuclear layer (INL) that contains the soma of bipolar and horizontal cells,

appear unimpacted by the loss of Sema3fa. As such, we could reasonably assess a role for Sema3fa in the larval CMZ without concern that developmental disruptions were responsible for CMZ phenotypes we observed in the *sema3fa^ca304* fish. Additionally, we took advantage of the fact that while *sema3fa* mRNA is restricted to the temporal eye embryonically [18], it is present in the CMZ of all eye quadrants by 72 hpf. As such, any CMZ-related phenotype that was present in the nasal CMZ would be unlikely to have arisen due to an embryonic defect.

Because the CMZ stem cell niche in the peripheral retina allows for ongoing growth of the zebrafish eye through adulthood, we asked whether eye growth was impacted in the absence of Sema3fa. Importantly, we reported previously that the size of the embryonic eye at 72 hpf was not impacted in the *sema3fa^ca304* allele [12]. To determine if post-embryonic growth beyond 72 hpf was impacted by Sema3fa loss, we measured the size of the left eye viewed dorsally at 10 dpf (Fig 2A, 2B and 2E) and laterally at 1 month (Fig 2C, 2D and 2F). When normalized to the nose to swim bladder length, there was no significant difference in eye area between the two genotypes at 10 dpf (Fig 2A, 2B and 2E), but the eye area normalized to head size was significantly smaller in juvenile mutant fish as compared to their WT counterparts (Fig 2C, 2D and 2F). These data suggest an eye growth defect that emerges as the mutant fish age. In support, the circumference of the 20 dpf eye, as indicated by the circumferential length of the IPL measured in frozen retinal sections, was significantly shorter in *sema3fa* mutant fish than WT (N = 4 independent experiments; WT 443.6 ± 48.1 μm (S.D.), n = 26 eyes; *sema3fa-/-* 369.9 ± 78.4 μm, n = 25 eyes; p = 0.0002 Mann Whitney U test). In summary, the smaller eye that develops over time in the *sema3fa* mutants points to a role for Sema3fa within the CMZ.

## Disruption of the organization of proliferating cells with *Sema3fa* loss

Alterations in proliferation could explain the reduced eye size in *sema3fa* mutants. To address this issue, we performed immunolabelling for Proliferating Cell Nuclear Antigen (PCNA) at 72 hpf; nuclei that stain uniformly and intensely for PCNA are in late G1 and early S-phase [24]. While most cells of the WT and *sema3fa* mutant CMZ expressed PCNA (Fig 2G, 2G″, 2H, 2H″ and 2I), the area of intense PCNA immunoreactivity normalized to the area of the CMZ was reduced in *sema3fa^ca304* fish as compared to WT (Fig 2H″ and 2J). Note that the CMZ was defined anatomically as the most peripheral region of the retina closest to the lens that extends until the edges of the inner and outer plexiform layers [25]. The elongated shape of progenitor cells further helped to demarcate the borders of the CMZ. To understand, if this reduction reflected changes in cycling progenitor numbers we performed EdU cumulative labelling [26,27], where 72 hpf larvae were incubated for progressively longer times between 1–12 hours in an EdU bath prior to using Click-iT to identify cells that were in or passed through S-phase over the course of the incubation period. These data were represented as a percentage of Hoescht-labelled cells in the CMZ that were Edu+ (Fig 2K). The number of CMZ cells that were in S-phase over an hour period, as normalized to the number of Hoescht-labelled cells in the CMZ, was reduced significantly in mutants vs. WT (Fig 2G′, 2H′ and 2K). This was also true for longer EdU labelling periods (Fig 2K). Indeed, the maximum labelling index (growth fraction, GF) as the proportion of cells that comprise the proliferating population of the CMZ, revealed by the plateau of the cumulative curve (Fig 2K), appeared reduced in the *sema3fa* mutants. This plateau was achieved in both genotypes by 4 hours (Fig 2K). The cumulative EdU labelling graphs allowed us to estimate the time when the maximum number of EdU+ cells were labelled (Tm), cell cycle length (Tc) and the length of S-phase (Ts) (slope = GF/Tc + Y; Tm = Tc-Ts). The first time point when the curve approached a slope of zero was taken as the Tm. To determine approximate cell cycle values, we fitted the first part of each curve, prior to reaching GF (1–4 hours), with a linear regression to identify the slope (WT, 0.719 = 4/Tc + .447, Tc = 15 hours, Ts = 11 hours; *sema3fa-/-* 0.686 + 4/Tc + .322, Tc = 11 hours, Ts = 7 hours; N = 2 independent replicates, n = 6–8 larvae/point). These data suggest that the cell cycle and time spent in S-phase is potentially shorter in the mutants. This data, combined with the observation that the maximum labelling index is smaller in the mutants, may indicate that the more rapidly dividing mutant progenitors leave the cell cycle to give rise to post-mitotic neurons sooner than their WT counterparts.

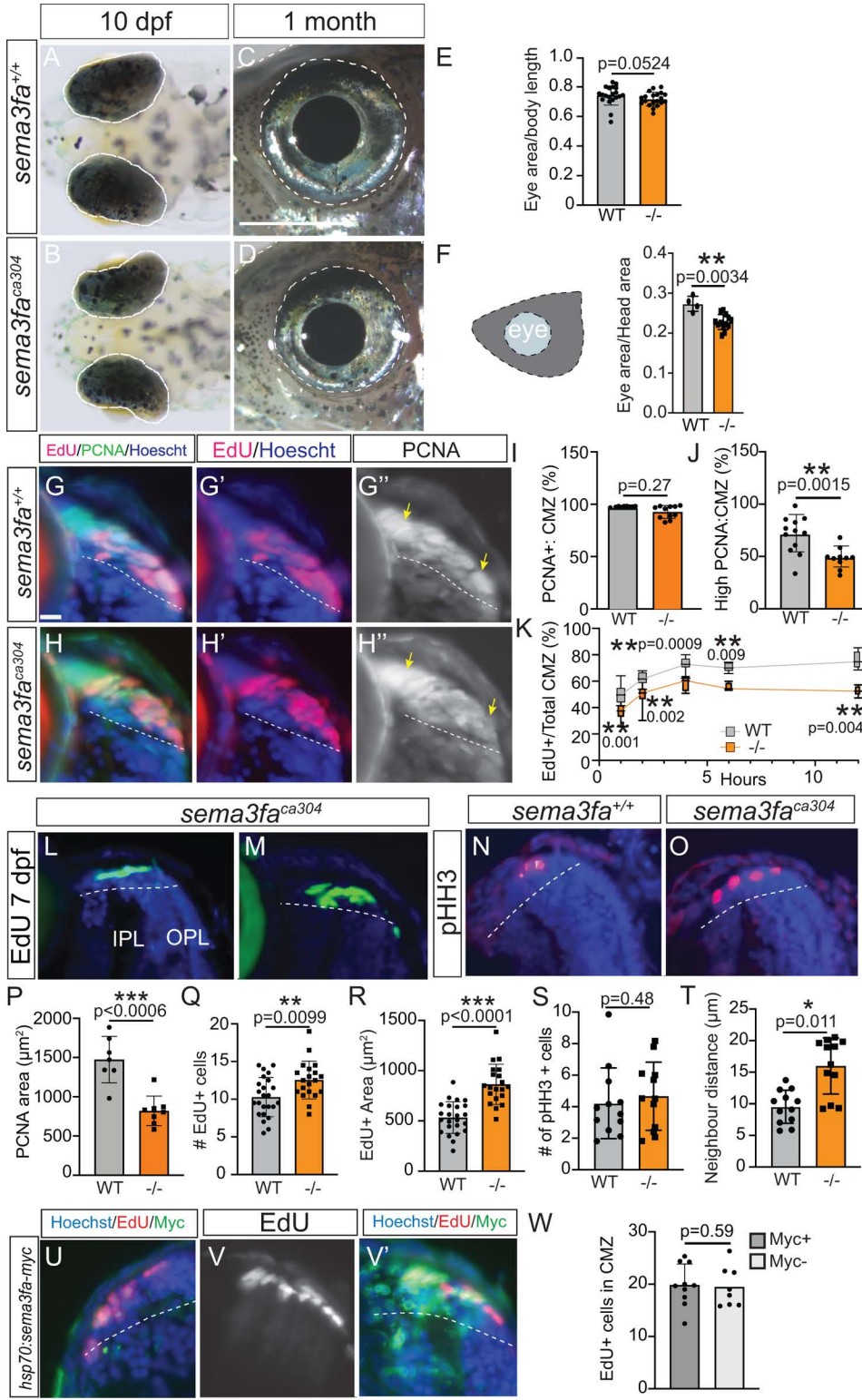

**Fig 2. Larval eye size and CMZ proliferating progenitors impacted in the *sema3faca304* mutant. A-D)** Dorsal view at 10 dpf (A, B) and lateral view at 1 month (C, D) of eyes (outlined in white dashed line) in WT (A, C) and *sema3fa^ca304^* (B, D) fish. **E-F)** Eye area normalized to nose to swim bladder length at 10-11 dpf (E) and to area of the head (measured from the nasal-most aspect of the head to the anterior boundary of the gills) at 1 month (F)

for WT and *sema3fa^ca304* fish. Pooled from N = 2 independent replicates. **G-K)** Cumulative EdU labelling of WT and *sema3fa^ca304* CMZ by bathing 72 hpf larvae in an EdU bath for 1-12 hours. One hour EdU labelling (G, G′, H, H′) of the CMZ in transverse retinal sections, also processed for PCNA immunolabelling (G, H, G′′, H′′) and for Hoescht to label nuclei (blue). The yellow arrows indicate uniform, intense PCNA immunolabel across the CMZ in WT (G′′) but lowered PCNA immunoreactivity in the mutant central CMZ (H′′). Graphs showing the percentage of Hoescht-labelled nuclei in the CMZ that are PCNA+ **(I)**, the area of intense PCNA expression normalized to CMZ area **(J)**, and the percentage of CMZ Hoescht-labelled cells that are EdU+ for each time point **(K)**. Data pooled from N = 2 independent replicates. **L-O)** Retinal cryosections of 7 dpf CMZ showing EdU-labelled progenitors after a 4-hour EdU pulse **(L, M)** and immunolabelling for pHH3 **(N, O)**. **P-T)** Graphs representing quantitation from retinal sections of the area of PCNA immunolabelling for the dorsal and ventral retina (P; N = 2 independent replicates), numbers of EdU+ cells (Q; N = 3 independent replicates) and the area of the CMZ they occupy (R; N = 3 independent replicates) from horizontal retinal sections, numbers of pHH3+ cells (S; N = 3) and the mean distance between them (T; N = 3) from horizontal retinal sections. **U-W)** Quantitation (W; N = 3 independent replicates) of the numbers of EdU+ progenitors (8 hour pulse) in Sema3fa-Myc-positive and Sema3fa-Myc-negative dorsal CMZ in transverse retinal sections of heat-shocked 5 dpf larvae injected at the one-cell stage with a construct with *sema3fa-myc* under the control of the *hsp70* promoter. The ventral (U) and dorsal (V, V′) CMZ of a single retinal section from a heat-shocked larvae, showing little (U) or Myc (V′; green) immunolabelling for Sema3fa-myc+ along with Hoechst (blue) and EdU (U, V, V′; red) label. For all graphs, values represent individual embryos, error is standard deviation, and Mann Whitney-U test was used to test statistical significance. IPL, inner plexiform layer; OPL, outer plexiform layer. Scale bar in C represents 500 μm for A, B and 1000 μm for C, D; scale bar in G is 10 μm for G-H, L-O and U-V.

With further development, the CMZ stabilizes and the numbers of mitotic progenitors in the retinal periphery decreases as the CMZ shrinks in size [15,28]. We examined if CMZ progenitor behaviour remained altered in the mutant fish. Similar to what we found at 72 hpf, at 7 dpf the area of the intense PCNA label was significantly reduced in *sema3fa* mutants as compared to WT (Fig 2P). Interestingly, when we performed a 4 hour EdU pulse at 7 dpf, which the cumulative labelling data (Fig 2K) suggests should catch most or not all dividing cells in the CMZ, we found slightly more S-phase labelled progenitors in *sema3fa* mutants than WT (Fig 2L, 2M and 2Q). The number of mitotically dividing cells in the late G2 and M phases of the cell cycle, as identified by phospho histone H3 (pHH3) immunolabelling of retinal sections (Fig 2N, 2O and 2S), however, was similar between the two genotypes. We noticed that the small numbers of CMZ EdU+ cells present at 7 dpf were distributed more broadly within both the nasal and temporal mutant CMZ than in WT (Fig 2L and 2M). Indeed, the average area of the CMZ occupied by EdU+ cells in *sema3fa^ca304* fish was increased significantly when compared to WT (Fig 2R). The same was true of pHH3 + cells, which were clustered in the WT CMZ but spread out within the mutant CMZ (Fig 2N and 2O), as quantitated by measuring the average distance between pHH3 + cells (Fig 2T).

To test whether Sema3fa directly controls progenitor proliferation we used a heat shock promoter to express a myc-tagged form of Sema3fa within the retina at 5 dpf to avoid disrupting embryonic retinogenesis. We injected a *hsp70:sema3fa:p3E-MTpA* Tol2 construct along with transposase mRNA into one-cell embryos, with expression of eGFP in the heart provided by *myl7:egfp* as the positive transgenesis marker. At 5 dpf, the embryos with green hearts were put at 38°C for 60 minutes to induce Sema3fa-myc protein expression. An 8-hour Edu pulse was performed 48 hours later, which should have caught all CMZ progenitors passing through S-phase. Note that this heat shock approach induced robust expression of Sema3a-myc protein in zebrafish embryos [29]. Regardless of whether there were no or significant numbers of Myc+ Sema3fa-expressing cells within the CMZ, even for the ventral (Fig 2U) and dorsal (Fig 2V′) CMZ of a single retinal section, similar numbers of EdU positive cells were present (Fig 2U, 2V and 2W). These data argue that Sema3fa is not sufficient to induce progenitor proliferation.

## CMZ area and cell numbers are unchanged by Sema3fa loss

To understand why proliferating progenitors are disrupted with the loss of Sema3fa we next asked whether the CMZ was grossly normal in the mutants. We first asked if the CMZ emerged normally during development by performing ISH for the retinal progenitor marker *vsx2* at 52 hpf [30] (Fig 3A and 3B). By this time point in most of both WT and mutants *vsx2* mRNA was present in the CMZ and mainly absent from the differentiating, central neural retina. These data argue that CMZ development is not delayed in *sema3fa* mutants. By 72 hpf, however, while *vsx2* mRNA was localized to the more distal WT CMZ, it was present more broadly within the *sema3fa^ca304* CMZ (Fig 3C and 3D). To quantitate this difference,

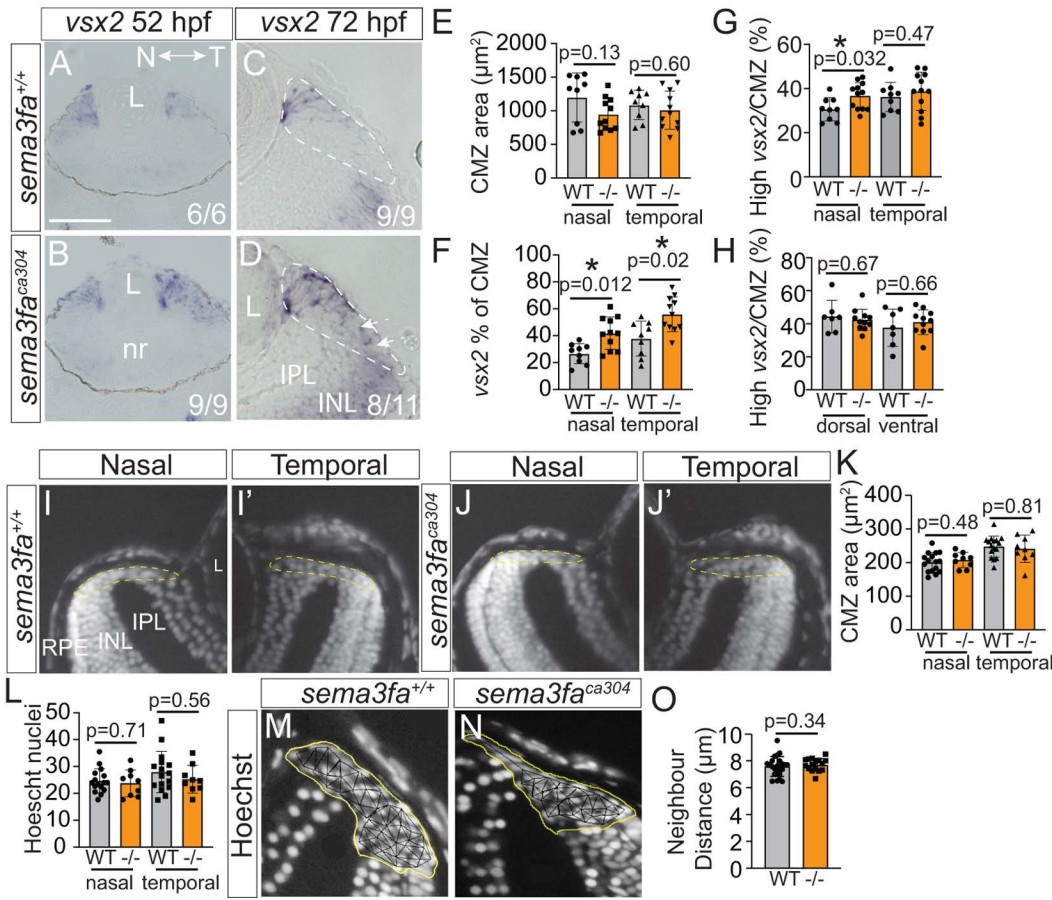

**Fig 3. Gross CMZ development and anatomy unimpacted by Sema3fa loss. A-D)** Retinal sections of WM ISH for *vsx2* that becomes restricted to the proliferative retinal periphery in WT (A) and *sema3fa^ca304* (B) fish at 52 hpf. At 72 hpf **(C-D)**, *vsx2* is expressed at high levels by distal progenitors of the CMZ in both genotypes, but is also expressed at low levels broadly through the mutant nasal CMZ (arrows). Note normal expression of *vsx2* by INL cells in both genotypes. n's represent numbers of fish with represented expression pattern. **E-H)** Quantitation of the *vsx2* expression domain within the CMZ. In sections through the central retina, the *vsx2* ISH expression domain and CMZ were outlined, using the edge of the inner and outer plexiform layers to mark the central limit of the CMZ; average CMZ area **(E)**, total *vsx2* area normalized to CMZ area **(F)**, and high *vsx2* expression area normalized to CMZ area **(G, H)**. **I-J)** Hoechst labelling of nuclei in horizontal cryostat retinal sections of 7 dpf WT (I) and *sema3fa^ca304* (J) nasal (I, J) and temporal (I', J') CMZ. **K-O)** Quantitative analysis of WT and *sema3fa^ca304* CMZ at 7 dpf from horizontal sections; **(K)** CMZ area; **(L)** Number of Hoechst-labelled nuclei; **(O)** Average distance between nearest neighbours, represented in M and N. INL, inner nuclear layer; IPL, inner plexiform layer; nr, neural retina; L, lens; RPE, retinal pigment epithelium. For all graphs, data is from N = 3 independent replicates, error bars are standard deviation, values represent individual embryos, and statistics represent Mann Whitney U-test. Scale bar in A is 150 µm for A, B and 40 µm for C, D, I-J, M, N.

we normalized the area of the *vsx2* expression domain within the CMZ to the area of the CMZ as measured in horizontal retinal sections. Importantly, the anatomical areas of both the nasal and temporal CMZ did not appear impacted substantially by the loss of Sema3fa (Fig 3E). The *vsx2* domain, however, made up a significantly larger portion of the *sema-3fa^ca304* CMZ than seen in WT (Fig 3F), suggesting possible alterations in the progenitors. Importantly, the *vsx2* domain expansion was evident in both the nasal and the temporal eye and so is unlikely to arise from defects in embryonic retinal development, as *sema3fa* mRNA is not present in the nasal embryonic retina [18]. Note that when we considered only the most highly expressing *vsx2* progenitors of the distal CMZ, there was only an increase in the *vsx2* area in nasal (Fig 3G), but not temporal, dorsal or ventral (Fig 3G and 3H) CMZ of mutants as compared to WT.

To determine whether the size of the CMZ and numbers of progenitors remained similar between genotypes with ongoing retinal growth, we assessed in a masked manner Hoechst-labelled nuclei in horizontal cryostat sections of WT and mutant 7 dpf eyes (Fig 3I and 3J). In the absence of Sema3fa, the average areas of the CMZ in both the nasal and temporal retina were not significantly different between WT and mutants (Fig 3K). In agreement, the total number of Hoechst-labelled nuclei within the CMZ was similar in the two genotypes (Fig 3L). Further, there was no significant difference in the average distance between nuclei in the WT and mutant CMZ (Fig 3M–3O). These data suggest that the numbers and packing of cells within the CMZ were not impacted by the loss of Sema3fa.

## Disruption in the expression of zone markers in the *sema3fa* mutant CMZ

The dispersion of proliferative markers (Fig 2) and the expansion of *vsx2* mRNA (Fig 3C and 3D) in the *sema3fa*ca304 CMZ led us to ask whether the spatial organization of the CMZ into transcriptional zones was impacted by the loss of Sema3fa. Previous reports suggest that the CMZ can be divided into four spatially discrete zones based on the unique expression of genes [31] (Fig 4C). We used the previous descriptions of gene expression within the CMZ of fish and frogs to characterize the zonal organization of the CMZ in the absence of Sema3fa [25,32,33]. These analyses were performed by WM ISH on 3 and 7 dpf zebrafish larvae, followed by plastic tissue sectioning to visualize and quantitate expression domains within the CMZ. The advantage of the ISH approach was that it allowed us to understand the spatial features of gene expression within the mutant CMZ. To compare the size of expression domains in retinal sections, we normalized the area of the gene expression domain to that of the CMZ in that retinal section.

Zone 1 is the distal region of the CMZ where retinal stem cells reside [25,32]. A marker for the stem cells of Zone 1 stem cells, *col15a1b* [34], was present in the peripheral-most region of the CMZ in both genotypes (Fig 4A, 4A′, 4B and 4B′). In the nasal CMZ, however, the *col15a1b* ISH domain was broadened in the mutants as compared to WT (Fig 4A′, 4B′ and 4F). In agreement, RT-qPCR analysis of cDNA collected from surgically isolated 72 hpf eyes indicated that *col15a1b* mRNA levels were upregulated significantly in the *sema3fa*ca304 retinas as compared to WT (Fig 4N). Both genotypes, however, expressed similarly a second marker of Zone 1, *bone morphogenetic protein 4* (*bmp4*), which is present in the distal portion of the embryonic mouse CMZ [35] (Fig 4D, 4E and 4G). In agreement, immunolabelling for polarity markers in the distal CMZ, the apical adherens junctions complex proteins Crumbs2a (Crb2a/zs4) [36,37] (Fig 4H and 4I) and atypical Protein Kinase C (aPKC) [38] (Fig 4J and 4K) appeared unaltered in Zone 1 of the CMZ. These data suggest that retinal stem cells are present in absence of Sema3fa, though some minor alteration of the nasal CMZ stem cell zone may be present.

Zone 2 was assessed by the expression of *ccnd1* [17]. At 3 dpf, the *ccnd1* domain in WT was compact and ovoid (Fig 4L), however, in *sema3fa*ca304 fish the domain's shape was irregular, with expressing cells located towards the central edge of the CMZ next to the neural retina (Fig 4M; white arrow). Further, cells appeared to show variable levels of *ccnd1* expression, with more centrally-located progenitors showing reduced *ccnd1* mRNA (Fig 4L and 4M; compare yellow arrows in M). We quantitated this expansion by measuring the areas of the CMZ and *ccnd1* mRNA domains and found that the proportion of both the nasal and temporal CMZ that expressed *ccnd1* was expanded significantly in mutants relative to WT fish (Fig 4O). While the area of the *ccnd1* domain was expanded in the mutants, suggesting disorganization of the domain, the levels of *ccnd1* mRNA were slightly decreased in dissected 72 hpf *sema3fa*ca304 vs. WT eyes, as assessed by RT-qPCR (Fig 4P), potentially reflecting progenitors expressing variable *ccnd1* mRNA levels. A disorganized *ccnd1* + Zone 2 was also observed in the mutant CMZ in the dorso-ventral axis; a greater portion of the mutant CMZ expressed *ccnd1* than seen in WT (Fig 4Q). Thus, Zone 2 was impacted in all quadrants of the mutant retina. Additionally, disruption of the *ccnd1* + progenitors remained as the eye matured, with a significantly larger *ccnd1* ISH domain normalized to the CMZ area present in the dorsal CMZ of 7 dpf mutants than WT (WT 34.4% ± 7.2%, n = 9 larvae; *sema3fa*ca304 51.6% ± 7%, n = 9 larvae; N = 2 independent replicates, p = 0.0002, Mann Whitney U test).

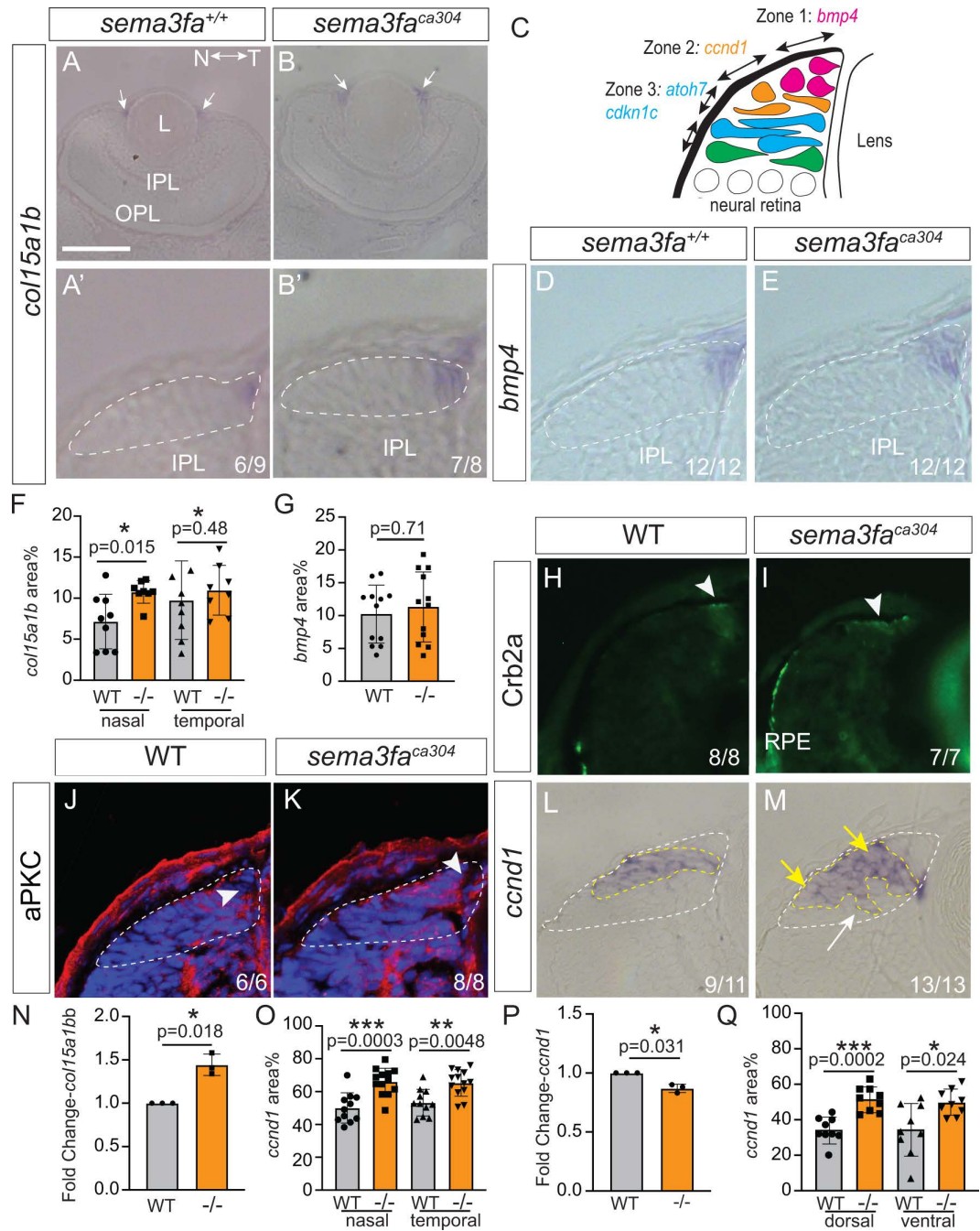

**Fig 4. Disruption of retinal progenitor cell domain with loss of Sema3fa. A-E)** Zone 1 markers *bmp4* (D, E) and *col15a1b*, at low (A, B) and high magnification (A′, B′), from plastic horizontal (A, B) and transverse (D, E) sections of WM ISH preparations. Schematic of the zonal structure of the CMZ **(C)**. **F-G)** Average *col15a1b* (F; N = 2 independent experimental replicates) and dorsal *bmp4* (G; N = 3 independent experimental replicates) expression domain areas normalized to CMZ area. **H-M)** Immunolabelling of distal CMZ cells for Crb2a/zs4 (H, I) and aPKC **(J, K)**, and plastic sections of WM *ccnd1* ISH **(L, M)**. Arrowheads point to label (H, I, J, K) in the distal CMZ that contains stem cells. n's represent numbers of embryos analysed that show WT-like expression **(H-L)**, and dispersed *ccnd1* expression (M; arrow). **N)** Fold change of *col15a1b* mRNA of *sema3fa*^ca304 relative to WT 72 hpf eyes. Each dot is an independent biological replicate. Paired two-tailed t-test performed on delta Ct values. **O)** Average normalized *ccnd1* ISH expression domain area for 72 hpf nasal/temporal CMZ in horizontal sections (O; N = 3 independent replicates). **P)** Fold change of *ccnd1* (Q) mRNAs of *sema3fa*^ca304 relative to WT 72 hpf eyes. Each dot is an independent biological replicate. Paired two-tailed t-test performed on delta Ct values. **Q)** Average normalized *ccnd1* ISH expression domain area in 72 hpf dorsal CMZ in transverse sections (N = 3 independent replicates). For graphs in F, G, O, Q points are

individual embryos, error is standard deviation, and statistics are Mann Whitney U-test. IPL, inner plexiform layer; L, lens; N, nasal; OPL, outer plexiform layer; RPE, retinal pigment epithelium; T, temporal. Scale bar in A is 100 µm for A, B; 30 µm for A′, B′, D, E, H-M.

We next assessed Zone 3, which contains cells that are leaving or left the cell cycle. Cells in both genotypes expressed mRNA for *cyclin dependent kinase 1c* (*cdkn1c*). Cdkn1c inhibits G1 Cyclin/Cdk complexes and is a negative regulator of cell proliferation, impairing movement of cells into S-phase of the cell cycle [39]. The mutant CMZ showed an irregularly shaped *cdkn1c* CMZ expression domain as compared to the compact domain of WT (Fig 5A and 5B), as reflected in significantly increased areas of *cdkn1c* expression in both the nasal and temporal CMZ of mutants (Fig 5C). Yet, similar to *ccnd1*, *cdkn1c* expression was slightly downregulated in the *sema3fa^{ca304}* 72 hpf eyes as compared to WT as assessed by RT-qPCR (Fig 5D). These data suggest that the cell cycle genes may be expressed at slightly lower levels in individual cells of the mutant CMZ, a phenotype not readily captured with the sections of WM ISH. Together with the *ccnd1* data our results suggest that the organization of the progenitors within the *sema3fa^{ca304}* CMZ is disrupted.

To further assess the organization of CMZ progenitors, we investigated the expression of transcription factors that turn on in cells of Zone 3 as they become post-mitotic and begin to differentiate. *atoh7* is a transcription factor of the basic helix-loop-helix family that is essential for the genesis of RGCs [40]. As described previously, at 72 hpf *atoh7* mRNA was restricted to the central region (Zone 3) of the WT CMZ [41]. Like *cdkn1c*, the *atoh7* expression domain in the *sema3fa^{ca304}* CMZ was irregular in shape (Fig 5E and 5F), and the proportion of the CMZ that expressed *atoh7* was increased significantly in the mutant vs. WT in both the nasal and temporal retina (Fig 5G). Additionally, a significant increase in *atoh7* mRNA level was observed by RT-qPCR (Fig 5H).

To corroborate the disorganization seen in *atoh7* expression in mutant CMZ at 72 hpf, we analyzed the expression of two additional CMZ Zone 3 markers. First, we assessed *hes6* mRNA expressed in differentiating cells of the central CMZ [42]. In lateral views, *hes6* mRNA was present in the peripheral portion of the CMZ adjacent to the lens in both the nasal and temporal retina of *sema3fa^{ca304}* but not WT fish (Fig 5I and 5J, yellow arrowheads). This expanded expression domain in mutants was evident in horizontal sections, where *hes6* was expressed by cells close to the lens (compare yellow lines, Fig 5I′ and 5J′). To quantitate the *hes6* phenotype we measured on lateral WM images the area of the *hes6* ISH that sat within the inner CMZ, ringed by the inner plexiform layer. This value was then normalized to the eye area for each embryo. A significant increase in the normalized *hes6* ISH area was seen in the *sema3fa* mutant vs. WT eyes (pooled data from N = 2 independent replicates; WT *hes6*:CMZ, 20.5% ± 6.0% (standard deviation), n = 25 larva; *sema3fa^{ca304}* 29.3% ± 7.0%, n = 25 larva; p < 0.0001, Mann Whitney U test). A similar expansion towards the peripheral CMZ was evident with *neurod4*, as we had noted previously [18], a transcription factor of the basic helix-loop-helix family that mediates neuronal differentiation as well as amacrine cell fate specification in the developing retina (Fig 5K and 5L; compare yellow lines) [43].

While Zone 2 and Zone 3 marker domains were expanded in the CMZ of *sema3fa^{ca304}* fish, we found the CMZ area was unchanged from WT (Fig 3E). Possibly in mutants there is some intermingling of the progenitors of the two zones. Alternatively, some progenitors may express markers of both zones. To address these possibilities, we first performed double FISH on WM WT and *sema3fa^{ca304}* 5 dpf fish for *ccnd1* (red), a Zone 2 marker, and *neurod4* (green), a Zone 3 marker. If progenitors intermingled, we expected mixed, but separate red and green FISH labels. In contrast, progenitors that expressed more than one zone marker would be yellow. Analysis of confocal images indicated that for the WT CMZ the Zone 2 and 3 markers were largely separate, though there was overlap of *neurod4* ISH signal with the central edge of the *ccnd1* domain (Fig 5M, arrow). The region of overlap was considerably expanded in the *sema3fa^{ca304}* CMZ (Fig 5N, arrow). We also performed double FISH for *ccnd1* and *atoh7* mRNA on transverse 72 hpf retinal sections. While these two genes are thought to lie in Zones 2 and 3, respectively, we found in the WT CMZ that *atoh7* mRNA sat within a small, centrally-biased region of the *ccnd1*-expressing Zone 2 (Fig 5O). Thus, certain previously characterized markers of Zone 3 progenitors may show some degree of overlap with progenitors of Zone 2. In the *sema3fa^{ca304}* CMZ, both *ccnd1*

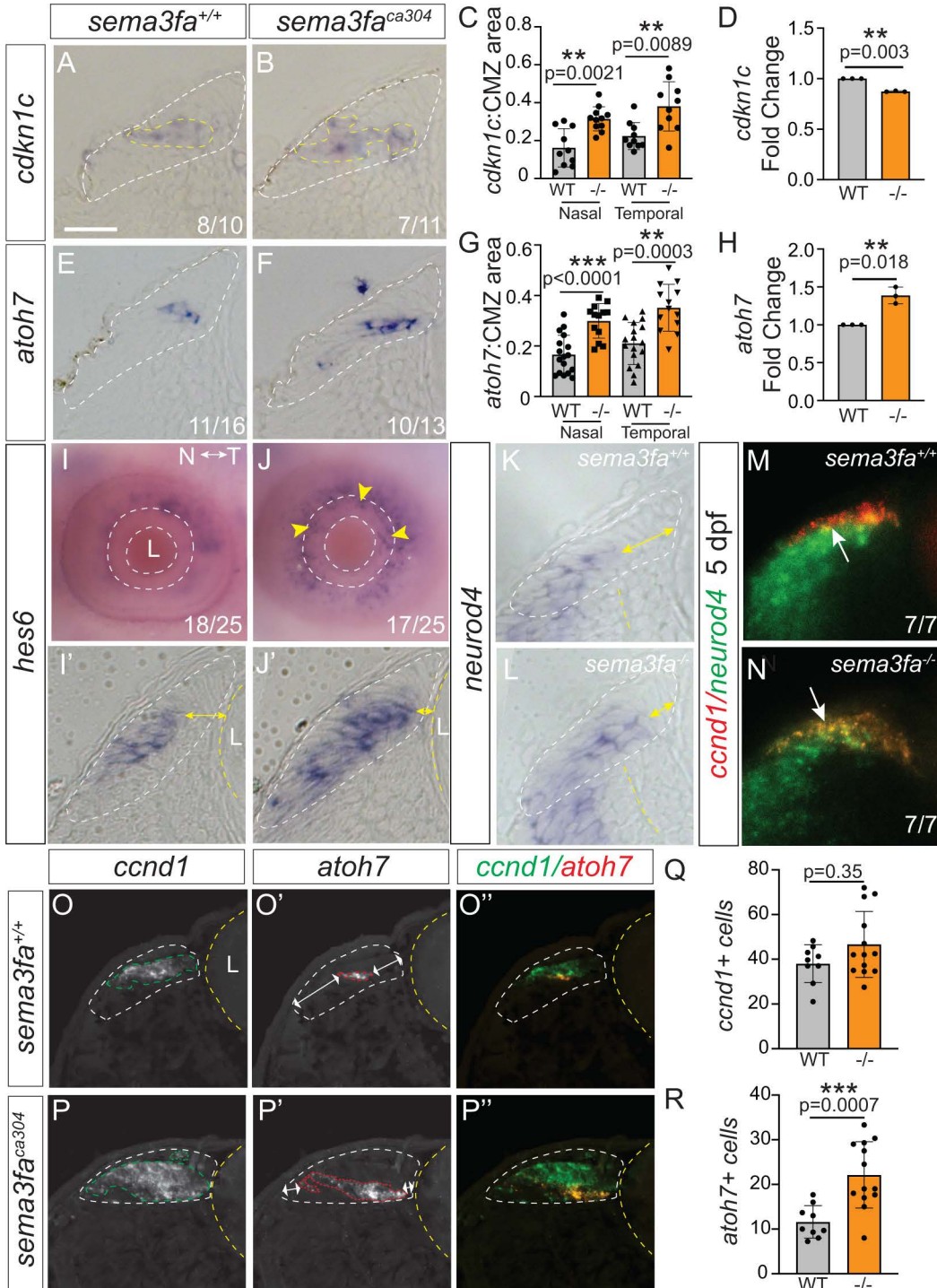

**Fig 5. Disrupted organization of committed CMZ progenitors in the absence of Sema3fa.** WM ISH for Zone 3 markers shown in 7 μm plastic sections (A, B, E, F, I′, J′, K, L), z-stack of confocal optical sections **(M, N)**, or lateral WM **(I, J)**. **A-H)** *cdkn1c* (A, B) and *atoh7* **(E, F)** ISH at 72 hpf, graphs of *cdkn1c* **(C)** (pooled data from N = 2 independent replicates) and *atoh7* **(G)** (pooled data from N = 4 independent replicates) mRNA domain areas normalized to CMZ area; Points for graphs are independent biological replicates, errors are standard deviation, and statistics are Mann Whitney U-test. Fold change of *cdkn1c* (D) and *atoh7* (H) mRNA levels in 72 hpf eyes relative to WT as measured by RT-qPCR (paired Student's t-test of delta CT values, with each data point an average of 3 technical replicates of RT-qPCR performed on mRNA isolated from n = 30 eyes). **I-J)** Zone 3 *hes6* marker in the 72 hpf eye in WM (I, J) and in plastic section (I′, J′). Expansion of *hes6 in situ* label to the most peripheral CMZ adjacent to the lens in *sema3fa^ca304*

eye (yellow arrows). n's refer to the numbers of eyes with significant (J) or little or no (I) *hes6* ISH label in the CMZ immediately adjacent to the lens. **K, L)** Plastic section of Zone 3 *neurod4* mRNA in the 72 hpf CMZ showing expansion (yellow arrows) of the *neurod4* label towards peripheral CMZ. **M, N)** Confocal images of 72 hpf CMZ processed for WM double FISH for *ccnd1* (red) and *neurod4* (green). There is more co-expression in mutant than WT (compare arrows in M and N). n's refer to the numbers of larvae showing WT levels of co-expression (M) and enhanced co-localization (N). **O, P)** Double FISH for *ccnd1* (O, P) and *atoh7* (O′, P′), and merge (O″, P″; green-*ccnd1*, red-*atoh7*) for WT (O) and mutant (P) 72 hpf CMZ. **Q-R)** Quantitation of Hoescht-labelled nuclei in *ccnd1*+ (Q) and *atoh7*+ (R) expression domains in transverse retinal sections processed for double FISH for *ccnd1* and *atoh7*. Points for graphs are individual larva, errors are standard deviation, and statistics are Mann Whitney U-test (N = 2 independent experiments). Scale bar in A is 20 µm for all panels except I, J (100 µm).

and *atoh7* mRNA domains were expanded (Fig 5P), as described earlier. The FISH approach allowed us to use Hoescht label nuclei to directly determine whether the increase in area was due to more cells expressing either of the two markers. We counted the number of nuclei in *ccnd1*- and *atoh7*- expressing domains (Fig 5Q and 5R). We found that *atoh7* appeared to be expressed by almost twice the number of cells (Fig 5R), but there no significant change in the numbers of *ccnd1*-expressing cells in the mutant as compared to WT (Fig 5Q). In both WT and mutants the *atoh7* mRNA domain sat within the *ccnd1*+ domain, though the *atoh7* mRNA signal occupied a larger extent of the *ccnd1*+ Zone 2 in the mutant fish (see double-headed arrows in Fig 5O′ and 5P′). These data suggest in mutants there is disrupted gene expression, where more *ccnd1*+ progenitors turn on Zone 3 markers such as *atoh7*, and as a result, progenitors that express *atoh7* or *neurod4* sit within regions of the CMZ they do not normally occupy. Thus, the loss of Sema3fa may alter the numbers of committed progenitors that express key genes that regulate the production of distinct neuron types.

### Mutant CMZ produce smaller radial cohorts of progeny

The disrupted spatial localization of mutant progenitors could impact neurogenesis by altering the extrinsic signals and cell-cell interactions encountered by a progenitor. To address this possibility, we investigated neurogenesis by EdU pulse chase labelling. We exposed 7 dpf WT and mutant fish to EdU for 4 hours, and let the fish grow for two weeks. Retinas were sectioned and processed for EdU, and the numbers and laminar identity of cells within individual radial cohorts of the central retina assessed (Fig 6A–6D). Of note, in larval retinas the only proliferating cells other than those of the CMZ are a small number of neurogenic Müller glia that provide new rod cells for the central retina [44]; lone rod progenitors within the central outer retina were not included in our CMZ analysis. Thus, most EdU-labelled cells in the retina adjacent to the CMZ at 20 dpf would have derived from CMZ progenitors. We collected 12 µm cryostat sections through the entire eye and counted, in a fashion masked to genotype, the number and layer distribution of EdU+ cells in each individual radially arranged cohort of cells in every second retinal section. We found that at 20 dpf, the average cohort size in the central retina was significantly smaller in mutants than in WT (Fig 6E), suggesting that any one mutant CMZ progenitor on average produces fewer retinal neurons than its WT counterpart. One possibility to explain the smaller cohort size is that newborn neurons die preferentially in *sema3fa* mutants. Yet, we found no significant difference in the mean number of TUNEL positive cells in 7 dpf eyes of WT vs. *sema3fa* mutants (WT 2.0 ± 1.5 (S.D) TUNEL+ cells, n = 10 embryos; *sema3fa-/-* 1.8 ± 1.1 TUNEL+ cells, n = 10; p = 0.78, Mann Whitney U test).

Retinal clones from both embryonic and CMZ progenitors can vary in size [15,31,45]. Yet, the distribution of progeny from multiple cohorts across the three retinal nuclear layers is comparable to the relative numbers of cells amongst the three layers; ~25% in the outer nuclear layer (ONL) that contains the photoreceptors, ~50% in the inner nuclear layer (INL) where the somas of interneurons and Müller glia reside, and ~25% in the RGC layer [45]. Note that we showed previously that *sema3fa* mutant retina exhibit normal lamination [12]. In investigating the layer distribution of cells (Fig 6F–6H), the cells of the RGC layer made up a greater percentage of the radial cohorts in the mutant retinas than was seen for WT (Fig 6H). Further, in comparing the cell composition of individual radially-oriented cell cohorts (as in Fig 6A and 6B) for a single representative WT and mutant eye, we observed more variability in the mutant vs. WT cohorts (Fig 6I and 6J). These data suggest that the distribution of labelled cells across laminae for an individual mutant cohort often did not prescribe

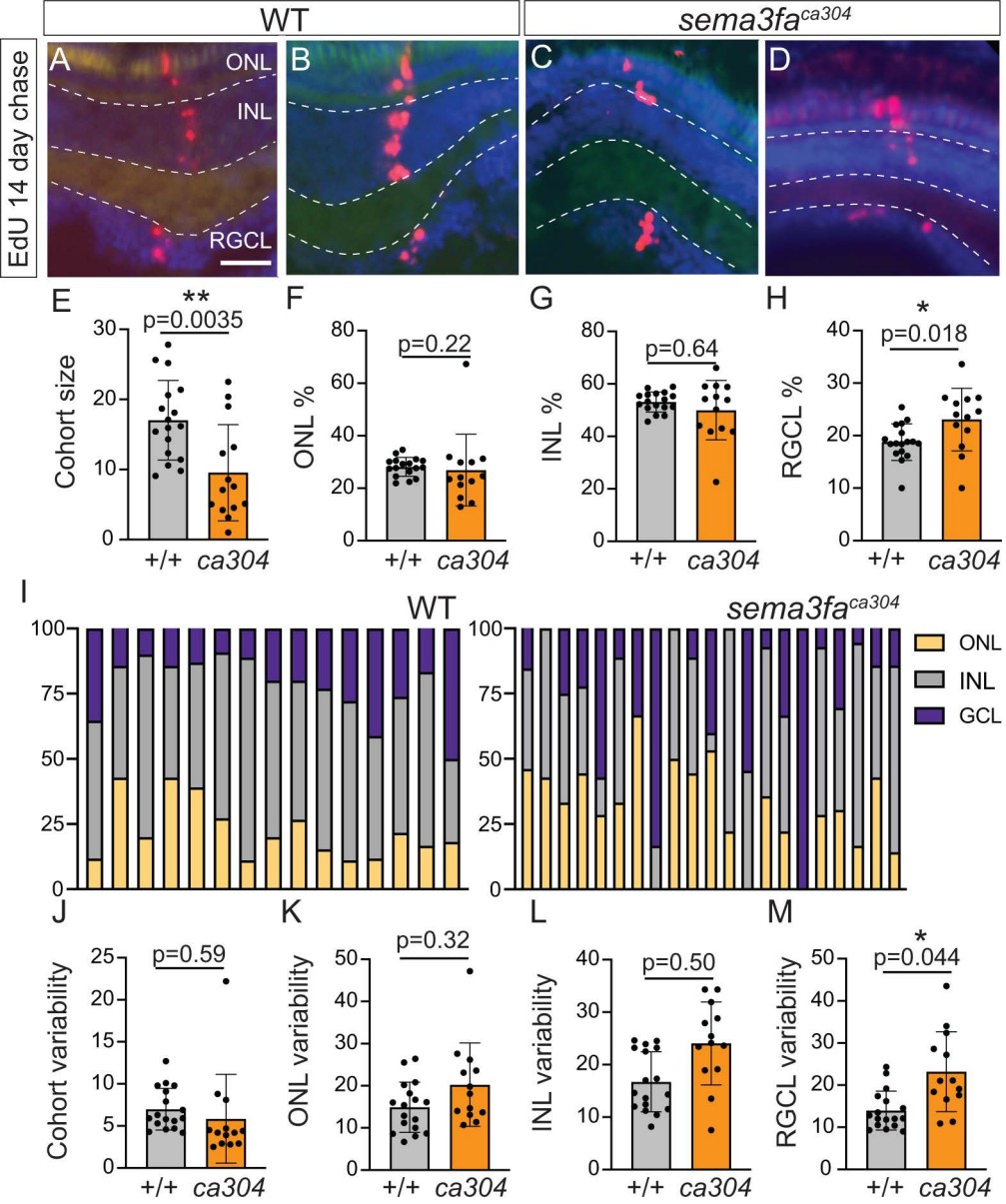

**Fig 6. Altered size and cell composition of CMZ-derived radial cohorts with loss of Sema3fa.** EdU labelling of retinas of 20 dpf zebrafish larvae incubated for 4 hrs with EdU at 5 dpf. **A-D)** Examples of radial cohorts of EdU-labelled cells in the central retina of separate WT (A, B) and *sema3fa^ca304* (C, D) fish. Panels are separate examples from different fish. **E)** Number of EdU+ cells (counted in every second section) in each radial cohort averaged per retina (WT, n = 17 larvae; mutant, n = 14 larvae; N = 2 independent replicates). **F-H)** The averaged distribution of EdU+ cells across the three major layers of the neural retina within individual radial cohorts of each larvae; outer nuclear layer (ONL) **(F)**, inner nuclear layer (INL) **(G)**, and retinal ganglion cell layer (RGCL) **(H)**. A larger percentage of cells of the radial cohorts reside within the RGC layer in the mutant as compared to the WT. **I)** Layer distribution of cells for each radial cohort of a representative WT and *sema3fa^ca304* retina. **J-M)** Percentages of cells in each retinal lamina were calculated for individual radial cohorts, and these percentages averaged for each retina (WT, n = 17 larvae; mutant, n = 14 larvae). The standard deviations for these averages are graphed in J-M. For all graphs, data points represent individual embryos, error bars are standard deviation. Statistical analyses use Mann Whitney U-test for E-H and a Levene's test for the equality of variance for J-M. INL, inner nuclear layer; ONL, outer nuclear layer; RGCL, retinal ganglion cell layer. Scale bar is 50 μm for A-D.

to the ~25:50:25 ratio that was typically seen in the WT fish. To test this possibility, we compared statistically the variance (standard deviation) of the average percentage distribution of cells in each of the three retinal layers for each eye (Fig 6K–6M) and found that the contribution of RGCs to individual cohorts was significantly more variable in mutants than WT (Fig 6M).

**In the absence of Sema3fa newborn retinal cells appear delayed in their exit from the CMZ**

Interestingly, the EdU-labelled radial cohorts were often closer to the CMZ in the mutant retinas than WT (Fig 7A and 7B). To quantitate this phenotype, we measured the distance between the peripheral-most edge of the CMZ and the lateral edge of each radially oriented EdU-labelled cohort in the 20 dpf retinas (Fig 7D; $d_1$). We found that in *sema3fa^{ca304}* fish the distance of the radial cohorts from the CMZ was reduced significantly (Fig 7C). To determine if newly born cells were slowed in their departure from the CMZ in the absence of Sema3fa, we measured the average distance of Edu-labelled progeny from the CMZ in central retinal sections after an 8-hour EdU pulse at 6 dpf and assessed 3 days later (Fig 7D; $d_2$). The leading edge of the EdU labelled cells was significantly further away from the peripheral edge of both the dorsal (Fig 7E) and ventral (Fig 7F) CMZ in WT vs. mutants. These data suggest that in the absence of Sema3fa newly born cells are delayed in their movement out of the CMZ. If true, one might expect the cells that express a potential receptor for Sema3fa, Plxna3 [14], would be found more distally within the *sema3fa^{ca304}* CMZ than observed in their WT counterparts. This is what we observed in plastic sections of 72 hpf retinas, where there was an expansion of *plxna3* label into more distal CMZ (Fig 7G and 7H; arrows). To quantify this phenotype, we measured the area of the *plxna3* ISH domain and normalized it to the area of the CMZ (Fig 7K). We found a significant increase in the area of the CMZ take up by *plxna3* ISH label in mutant as compared to WT CMZ. To confirm these data, we performed double FISH for the Zone 2 marker *ccnd1* and *plxna3* on retinal sections from WT and *sema3fa^{ca304}* fish at 72 hpf. In this way we could compare *plxna3* ISH label to that of a more peripherally expressed CMZ marker. In WT sections, there was a clear separation of the *ccnd1* and *plxna3* expression domains, with the *plxna3* domain located more towards the central retina (Fig 7I). In contrast, in the mutant CMZ *plxna3* cells sat immediately adjacent and within the *ccnd1* domain (Fig 7J), arguing that Sema3fa-responding cells fail to move robustly out of the CMZ and that potentially progenitors begin to express a receptor normally expressed by centrally-located CMZ cells that have left the cell cycle.

A possible explanation for the disorganization of the CMZ in *sema3fa* mutants is that normally Sema3fa negatively regulates the expression by progenitors of cell adhesion molecules. Indeed, SEMA3F inhibits E-cadherin mediated cell adhesion of cultured breast cancer cells [46]. In this scenario, by keeping the levels of adhesion molecules on progenitors low, Sema3fa would help the progenitors move smoothly through the CMZ in a distal to central direction. To explore whether adhesive signaling was potentially upregulated in mutant CMZ progenitors we performed immunolabelling for the adhesive signaling proteins ß-catenin and N-Cadherin (Cdh2) (Fig 7L–7O). ß-catenin controls the adhesion of retinal precursor cells in the embryonic mouse retina [47] and is required to maintain proliferation of embryonic zebrafish retinal progenitors [48], while zebrafish CMZ progenitors express Cdh2 [25]. We directly assessed protein expression within the CMZ by immunolabelling, rather than performing Western blot analyses of surgically isolated eyes, as both proteins are also expressed in the neural retina. Interestingly, we found both proteins appeared upregulated within the mutant CMZ as compared to WT, with an expansion of the immunolabelling to more central regions of the CMZ (Fig 7L–7O; compare arrows). Indeed, a greater percentage of the CMZ exhibited robust fluorescent label for Cdh2 in the absence of Sema3fa as compared to WT (Fig 7P).

To test this model further, we turned to the heat shock approach described earlier, which results in mosaic expression of Sema3fa-myc in the CMZ, which we verified by Myc immunolabelling. We predicted that overexpression of Sema3fa-myc would cause cells adjacent to Sema3fa-myc+ cells to downregulate expression of Cdh2. Indeed, while Cdh2 appeared evenly distributed in the more distal-central CMZ of injected fish that showed no Myc-immunolabelling within the CMZ upon heat shock (Fig 7Q), in CMZ with Myc+ cells, Cdh2 immunolabelling was uneven and showed gaps

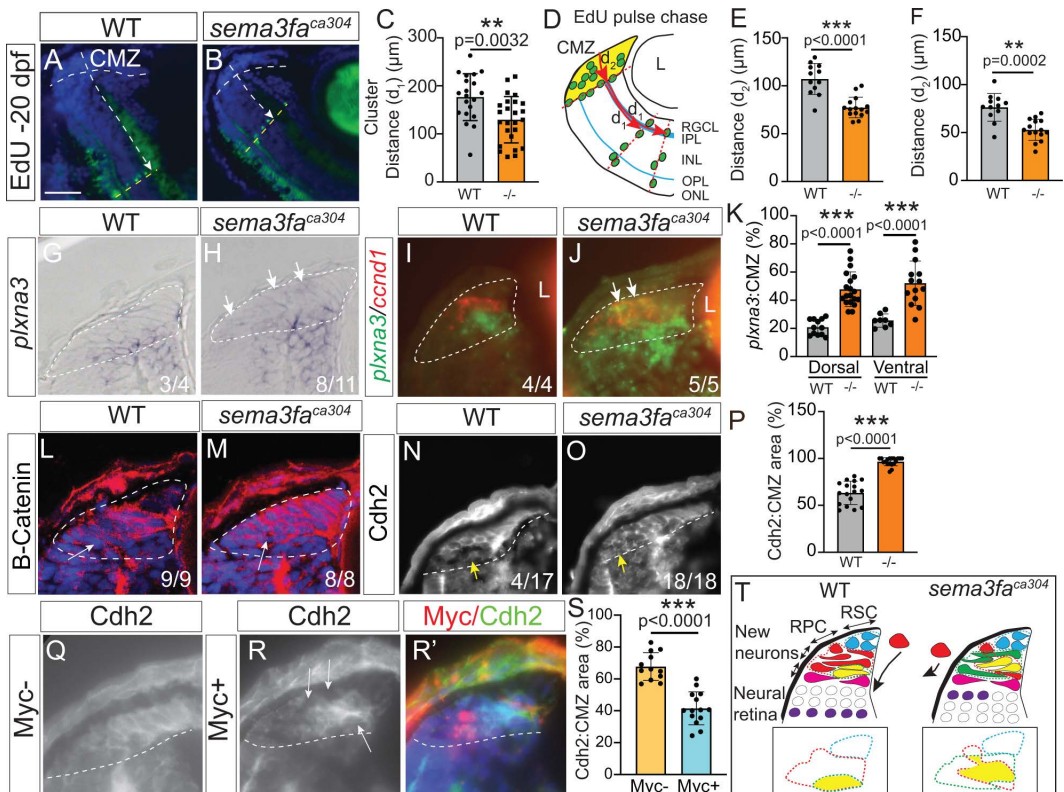

**Fig 7. Sema3fa may promote smooth movement of progenitors through CMZ by downregulating cell adhesion. A-B)** Radial EdU+ cohorts (green) that have moved away (dotted arrows) from the CMZ in WT (A) and *sema3fa^ca304* retina **(B)**. A 4 hour EdU pulse was performed at 7 dpf and retinal sections from larvae fixed at 20 dpf processed for EdU labelled cells. **C)** Average distance ($d_1$; D) of EdU+ radial cohorts from the CMZ in *sema3fa^ca304* is reduced significantly as compared to WT. N = 3 independent replicates. **D)** Schematic showing measurement of the distance ($d_1$) moved from the central CMZ edge by radial cohorts of EdU+ cells (green) 2 weeks post-EdU label, and the distance ($d_2$) moved from the peripheral CMZ edge by the central-most cohort of EdU+ cells (green) 3 days post-EdU label. **E-F)** Graphs showing the average distance ($d_2$; D) of each radial cluster of EdU-labelled CMZ-derived cells from the peripheral-most CMZ, as measured in 9 dpf dorsal (E) and ventral (F) retina. N = 2-3 independent replicates. **G-H)** Retinal sections of the CMZ from WM ISH for *plxna3* mRNA in WT (G) and *sema3fa^ca304* (H) 72 hpf fish. n's represent the numbers of larva showing either WT (G) or expanded (H; arrows) *plxna3* expression. **I-J)** WT (I) and *sema3fa^ca304* (J) CMZ processed by double FISH for *plxna3* (green) and *ccnd1* (red) mRNA. *plxna3*-expressing CMZ cells are found ectopically (arrows) in the peripheral CMZ and co-express *ccnd1* (yellow). n's represent the number of individual embryos that exhibit separated (I) or overlapped (J) *ccnd1/plxna3* expression domains. **K)** Quantitation of *plxna3* expression domain normalized to CMZ area of 72 hpf horizontal retinal sections (N = 3 independent replicates). **L-O)** Immunolabel for ß-catenin (L, M) and Cdh2 (N, O) in the 72 hpf CMZ. Arrows indicate reduced immunolabel of the central CMZ in WT but not mutant. n's indicate the numbers of fish showing either preferential, bright label of the more distal CMZ (L, N) or immunolabel distributed across both the distal and central CMZ (M, O). **P)** Quantitation of the percentage of the CMZ exhibiting robust Cdh2 immunostaining fluorescent label. Data from N = 4 independent replicates. **Q-S)** Cdh2 immunolabelling of the CMZ in larvae injected at the 1-cell stage with a *hsp70:sema3fa-myc* construct and heat-shocked at 38°C for 60 minutes at 52 hpf and fixed 24 hours later. Quantitation (S) of the domain of high Cdh2 expression as a percentage of CMZ area in CMZ exhibiting Myc+ cells (R, R'; red) and no Myc+ cells (Q). Data from N = 3 independent replicates. For all graphs, dots represent individual fish, error is standard deviation, and Mann Whitney U-test was performed for statistical analysis. **T)** Schematic of the CMZ of WT and *sema3fa^ca304* fish showing normal retinal stem cell (RSC) localization to the CMZ periphery, but disrupted organization of committed and proliferating (retinal progenitor cells; RPC) in the mutants. Fewer neurons (purple) are produced and move less distance from the CMZ in mutants. We propose a model whereby Sema3fa negatively regulates cell adhesion molecule expression by progenitors. As such, *sema3fa^ca304* CMZ progenitors are more adhesive and do not move smoothly through the CMZ (red cells, arrows). Boxes show impact on the zonal organization of the CMZ. Scale bar in A is 100 μm for A, B and 25 μm for G-J, L-O, Q-R.

in expression (Fig 7R and 7R'; arrows). Indeed, the area of strong Cdh2 expression was decreased in CMZ that showed Myc+ cells as compared to the non-Myc expressing CMZ (Fig 7S). These data support the idea that Sema3fa negatively regulates adhesion between CMZ progenitors.

## Discussion

We present evidence that Sema3fa ensures appropriate neurogenesis by situating progenitors within the post-embryonic zebrafish CMZ. *sema3fa* along with its high-affinity receptors are expressed within this neurogenic niche. Sema3fa loss disrupts the normal organization of the CMZ into compact domains of transcriptionally and functionally distinct progenitors (Fig 7T). Further, cell cycle dynamics and neurogenesis are disturbed; with individual EdU-labelled radial cohorts derived from the mutant CMZ smaller in size and altered in composition from those seen in WT. We propose these alterations explain the smaller size of mutant vs. WT juvenile eyes. These data suggest that the zonal organization of CMZ progenitors is important for appropriate retinal neurogenesis, a feature that requires an extrinsic factor secreted by the progenitors themselves, Sema3fa.

CMZ progenitors likely respond to Sema3fa as they express mRNAs for Nrps/Plxns that exhibit high Sema3F affinity [13,14]. The fact that the juvenile CMZ continues to express ligand and receptor mRNAs argues for ongoing Sema3fa signalling in progenitors. Of note, Sema3fa is expressed in the CMZ of *Xenopus laevis*, indicating a potential conserved role for Sema3f in regulating CMZ neurogenesis [49]. While rodents and humans lose a CMZ postnatally [5,50], the embryonic rat retina expresses mRNA for *Sema3f* as well as its receptors [51], suggesting a role for SEMA3F in rodent retinal development.

The zebrafish CMZ serves to sustain eye growth through the life span [52,53]. The small eye size of one-month old mutant fish is unlikely to arise from developmental delay or impaired embryonic eye growth, as their eyes as early larvae are normal in size. Instead, we propose the smaller size of radial cohorts that emerge from the mutant CMZ explains the reduced eye size. Thus, Sema3fa within the larval CMZ appears to be important for continued eye growth, and, given the expression of *sema3fa* and *nrp2b* in the month old CMZ, through into the adult. Future work can address whether Sema3fa plays a role in retinal regeneration mediated by the CMZ [54], given the apparent ongoing expression of *sema3fa* mRNA in this niche.

We propose that Sema3fa supports eye growth via maintenance of the zonal organization of the CMZ (Fig 7T). This organization is disrupted in the absence of Sema3fa. Expression domains of proliferating (*ccnd1*) and committed (*atoh7*, *neurod4*, *hes6*) progenitor markers are compact in WT but expanded and distorted in the mutant. In agreement, in the maturing CMZ, pHH3 and EdU labelled mutant progenitors are scattered rather than clustered as in WT. The peripheral CMZ (Zone 1), which contains the stem cells [5], appears less impacted by the loss of Sema3fa: While *col15a1b* mRNA is upregulated with a larger expression domain in the nasal CMZ, *bmp4*, aPKC and Crb2a label within this domain appear unchanged. These observations suggest that Sema3fa signaling primarily acts on central CMZ progenitors. In support, *sema3fa* receptor mRNAs are largely absent from the peripheral CMZ. Why do the markers of proliferating and committed progenitors make up a larger region of the mutant CMZ? Our data argue against the idea that there are more progenitor cells. In fact, the cumulative EdU labelling data argue there are fewer cycling progenitors in the mutants, at least at 72 hpf. Further, the size of the CMZ is comparable between the two genotypes. Instead, the FISH co-labelling of zone markers suggests that more progenitors express both Zone 2 and Zone 3 markers, indicating alterations in the transcriptome of individual progenitors.

We propose that progenitors in the central CMZ secrete Sema3fa into the extracellular milieu to promote zonal organization, and that with loss of Sema3fa the changes in expression of the different progenitor markers arise secondary to defects in this spatially organized niche. Indeed, Sema3fa has no direct effect on progenitor proliferation when overexpressed within the CMZ. Initially identified as axon guidance molecules, secreted SEMA3s also regulate progenitors [55]. Sema3fa may act, as secreted Sema3s are known to do for migrating neurons and axons, by directly situating progenitors through attractive or repulsive mechanisms [7]. For instance, Sema3fa could attract "like" CMZ progenitors to a specific zone, in that SEMA3F attracts oligodendrocyte precursor cells [56]. Yet, *sema3fa* mRNA appears too broadly expressed within the CMZ for such a role, and we find increases in adhesive molecules, Cdh2 and ß-catenin, when Sema3fa is absent.

Alternatively, Sema3fa could act via Nrp-Plxn to repel progenitors and move them through the CMZ towards the neural retina. Indeed, SEMA3F can serve as a repulsive guidance molecule [55,57,58]. This centralwards push would be lost in mutants, resulting in disorganization of the niche. We do find the movement of newly generated neurons out of the CMZ and into the neural retina is impaired; in *sema3fa* mutants EdU-labelled radial cohorts are closer to the CMZ and *plxna3*-expressing progenitors are more widely distributed. Yet our data support an alternative model, where Sema3fa limits adhesion between cells, allowing for smooth distal-to-central progenitor movement within the CMZ. Upregulation of cell-cell adhesion upon Sema3fa loss would produce "sticky" progenitors that would exhibit hampered motility (Fig 7T). Indeed, SEMA3F uses NRP1 to inhibit E-cadherin and ß-catenin mediated adhesion of breast cancer cells [46]. Our evidence supports this notion, with both ß-catenin and Cdh2 protein levels upregulated in the mutant central CMZ as compared to WT, and the Cdh2 domain reduced in CMZ that overexpress Sema3fa. Our working model is that through adhesive mechanisms Sema3fa ensures proper spatial localization of CMZ progenitors.

Interestingly, SEMA3F plays a de-adhesion role in forebrain neurogenesis, acting via NRP1B to support neurogenesis by promoting the detachment of the apical endfoot of cortical progenitors from the ventricular surface: Loss of SEMA3F/NRP1 results in decreased neurogenesis, and SEMA3F-Fc inhibits cortical progenitor adhesion to NRP1-Fc coated coverslips [11]. Several differences between the CMZ and cortical systems suggest potentially distinct roles for SEMA3F. First, in SEMA3F mouse knockouts, pHH3+cells remain close to the ventricular surface and to each other [11], while pHH3+cells are further apart in the zebrafish *sema3fa* mutant CMZ. Second, our data suggest a CMZ autonomous role for Sema3fa; Sema3fa secreted by CMZ progenitors acts via Plxn/Nrp expressed by CMZ cells. In contrast, SEMA3F is made by the choroid plexus and provided asymmetrically to the endfeet of the cortical progenitors via the cerebrospinal fluid [11]. Finally, ß-catenin and Cdh2 levels are unaltered in the SEMA3F KO cortex. Thus, the particular context for SEMA3F, related both to its spatial presentation and receptor use (e.g., Nrp1 vs. Nrp2), may mean that SEMA3F plays different roles in regulating progenitor biology in distinct stem cell niches. Understanding where Sema3fa protein is localized in the zebrafish CMZ will provide some insight as to its role, however, our zebrafish Sema3fa antibody works well in Westerns but not immunostaining [12].

Why progenitor disorganization disrupts CMZ neurogenesis, with respect radial cohort size and variability, remains to be determined. Our TUNEL analysis at 7 dpf argues that retinal apoptosis is not a contributory factor. The cumulative Edu data suggest an altered cell cycle may play a role. Specifically, we find in the early mutant CMZ there are fewer total numbers of cycling cells than in WT. The cycling cells appear to exhibit a shorter S-phase and cell cycle. In support, we find decreased numbers of high PCNA-expressing mutant progenitors that sit at the G1-S transition [24], and lowered *ccnd1* mRNA, noting that during the normal cell cycle Ccnd1 levels drop to allow cells to move into S-phase [59]. The fact that high-expressing PCNA+ progenitors are also reduced in the 7 dpf mutant CMZ indicate that the shortened cell cycle may persist. Why then are there more EdU+ cells in the mutant CMZ at 7 dpf? One possibility is that the maturation and stabilization of the CMZ that occurs over early larval development is delayed in the mutant, and that the normal reduction in dividing cells seen by 7 dpf has not yet occurred [15,28]. Alternatively, progenitor numbers could be similar between genotypes at 7 dpf, and yet if the cell cycle is shorter in mutants we might expect to catch more cells moving into and through S-phase in the mutants, which would be seen as more EdU+ cells. Such an increase would have been masked by the reduced growth fraction in the mutants at 72 hpf. Further analysis of cell cycle kinetics over the larval period and into the adult will determine how the loss of Sema3fa impacts the CMZ progenitor over the life span. Interestingly, progenitors committed to neuron production rather than expansion exhibit a shorter S-phase [60]. Potentially, *sema3fa* loss pushes progenitors to symmetric neuronal divisions resulting in fewer non-neurogenic progenitor divisions that expand the progenitor pool, a scenario that would ultimately result in less neuron production.

The post-embryonic CMZ produces retinal clones that contain a similar cell-type composition (~25% ONL, ~50% INL, ~25% RGCL) to those derived from embryonic progenitors [15,45]. Interestingly, mutant radial cohorts exhibited an overrepresentation of RGCs and were generally more variable in composition. The former may reflect an increase in the

numbers of *atoh7*-expressing CMZ progenitors. Sema3fa may indirectly regulate neurogenesis by governing progenitor location and the specific set of extrinsic molecules encountered by each progenitor. Indeed, post-mitotic cells along the central edge of the CMZ provide pro-differentiation feedback signals to CMZ progenitors [17,61]. Mislocalized progenitors would encounter a different set of differentiation signals, changing their transcriptome and/or the probability of symmetric vs. asymmetric type divisions [62], and alter both clone size and composition [5,62].

The expression by zebrafish CMZ cells of mRNAs for several Sema3 receptors, but with reported differential affinity for SEMA3F, suggests that different receptors may mediate distinct Sema3F roles and/or the actions of additional Sema3 family members. For instance, *sema3gb* mRNA is present in the 72 hpf zebrafish CMZ [16]. Alternatively, the Sema3s could be provided by cells outside the CMZ, as is seen for SEMA3B that is made by the choroid plexus and regulates neurogenesis in the embryonic mouse cortex [11]. Indeed, in zebrafish a subset of INL cells express *sema3aa* and *sema3ga* is expressed by RGCs [16]. Finally, blood vessels could provide Sema3s to the adjacent CMZ progenitors [15] as they do for growth factor support of other stem cells [25,63]. Different sources of SEMA3s, distinct involvement of SEMA3s, and different receptor involvement all allow for a greater diversity of regulation of neurogenesis.

## Materials and methods

### Ethics statement

All animal protocols were performed ethically as approved by the University of Calgary Animal Care Committee (ACC Certification AC23-0127).

### Zebrafish husbandry

Zebrafish (*Danio rerio*) were housed according to standard conditions on a 14-hour light/10-hour dark cycle at 28°C. We incrossed wild type (WT) Tupfel Long Fin (TL) fish, *sema3fa*$^{ca304/ca304}$ fish, and for the latter their corresponding WT line produced from the same clutch used to generate the specific *sema3fa*$^{ca304/ca304}$ fish line [12]. Their embryos were raised in E3 medium supplemented with 0.25 mg/L methylene blue and staged by convention [64,65]. For certain experiments, pigment synthesis was inhibited by adding 0.003% (w/v) 1-phenyl-2-thiourea (PTU) to the E3 solution at 12–24 hpf.

**Eye size measurements.** Live *sema3fa*$^{ca304}$ and WT siblings were imaged from a dorsal orientation at 10 days post fertilization (dpf). Individual fish were placed in E3 supplemented with 164 mg/L of MS-222 and imaged using a Zeiss Axiocam HRc stereoscope. Separate sets of one-month old fish were imaged from a lateral orientation. The areas of the pigmented eyes of larvae (dorsal view) and juveniles (lateral view) were measured in a masked fashion using Axiovision software and normalized to the nose to swim bladder length and the size of the head (measured to the back of the gills), respectively.

**Nearest neighbour analysis.** For Hoechst-labelled nuclei within the CMZ of retinal sections of 7 dpf fish, we used the Delaunay Triangulation plugin available in Fiji [66] to determine distances between nuclei of nearest neighbours. The CMZ domain was defined by the anatomical landmarks of the edges of the inner and outer plexiform layers. The automated multi-point cell selection within FIJI was performed by using a specified ROI-based on pixel threshold allowing for the determination of the center of all the CMZ nuclei. The central and distal CMZ were analysed individually to avoid edge effects. Delauney triangulation was then used to determine the average distance between cells.

**Real time quantitative PCR (RT-qPCR).** Total RNA was extracted from thirty 72 hpf eyes and RT-qPCR was performed as per [18] (N = 3 independent replicates). Primers include *ß-actin*, *bmp4* F: GACCCGTTTTACCGTCTTCA & R: TTTGTCGAGAGGTGATGCAG, *ccnd1* F: ACAGCAACCTGTTGAATGAC & R: GGCCAGATCCCACTTCAGTT, *cdkn1c* F: CGCCGCAAATTACAGACTTC & R: ATGTGCCGGCTTGAAGGTAA, *col15a1b* F: GGCAAAGATGGCAGAGATGG & R: CTCACCCTTTGCTCCTTTCG.

**In situ hybridization.** Digoxigenin (DIG) or 2,4-Dinitrophenol (DNP) labelled RNA probes were generated from plasmid templates or by PCR-amplification; the 5′ end of the reverse primers contained the DNA sequence for either

SP6 or T7 RNA polymerase (SP6: 5′-GCATTTAGGTGACACTATAGA-3′ or T7: 5′-GAAATTAATACGACTCACTATAGG-3′) as described previously [67]. Probes included: *sema3fa*, *nrp2a*, *nrp2b*, *plxna1a*, *plxna1b*, *plxna2*, *plxna4* [12,21,68,69], *atoh7*, *neurod4* and *vsx2* [18], *plxna3* (pCRII, linearized with EcoRV); *col15a1b* F: ACACTCTGGAAGTGTTGATATC, R: GGTCTGGCACTTTAAAAGACGCA, *cdkn1c* F: CTCTGTTGGCGGAGCCATTA, R: GAGATTACGAGTGGGAGGCG, *ccnd1* F: CACTTCCTTGCCAAACTGCC, R: GCATAGAAGGGGTGGCCAAGG, *hes6* F: AAAGACGAGGAAACCGCTGG, R: GCCTGATCTCCGGCATCTACC. *bmp4* F: CTGCCAGGACCACGTAACAT, *bmp4* R: GTGGCGCCTTTAACACCTCAT. *nrp1a* F: CGATCAGGGAGATGAGCACC, *nrp1a* R; CGTCTACAGCGATGTCTCCC. *nrp1b* F: AGGTGTTTGATGGTGCGGAT, *nrp1b* R: CGACGATGCAGCTATCCTGT.

RNA whole mount (WM) ISH was conducted using standard protocols [67]. For double fluorescent ISH (FISH) in WM, a similar protocol was followed except samples were stained 30 minutes in the dark in fluorescein tyramide or Cy3 tyramide at 1:50 in 1x Amplification Buffer (Akoya BioSciences) containing 2% dextran sulfate. For double FISH on frozen sections, the same protocol as for WM double FISH was followed except the RNA probes were added directly to 12 µm cryostat sections on glass slides, and slides were coverslipped and incubated at 65ºC overnight in probe.

**Immunohistochemistry.** Immunohistochemistry was performed following established protocols on 12 µm cryostat cut frozen retinal sections [18]. Slides were incubated overnight with primary antibody diluted in block (1% Bovine Serum Albumin (BSA)/10% Normal Sheep Serum (NSS) in PBT) at 4°C: Phosphohistone H3 (pHH3; 1:250, Millipore), Proliferating Cell Nuclear Antigen (PCNA; 1:100, Novus Biologicals), atypical PKC (aPKC (PKC epsilon (H-1)); 1:500, Santa Cruz), Crumbs2a (Crb2a/zs4; 1:75, ZIRC), c-Myc (9E10, 1:500, Novus Biologicals), ß-catenin (1:500, BD Transduction). Samples were washed with PBT and incubated in secondary antibody (Alexa Fluor 488/555; anti-mouse/anti-rabbit, 1:1000, Molecular Probes) supplemented with 2.5 µg/mL Hoechst (Invitrogen) for 45 minutes at room temperature prior to mounting with Aquapolymount.

**Click-iT EdU imaging.** The Invitrogen Click-iT TM 5-Ethynl-2'-deoxyuridine (EdU) Cell Proliferation Kit (Catalog #C10338) was used to label proliferating cells in retinal cryosections according to the manufacturer's guidelines. Larvae at the indicated ages were bathed in 250–500 µM Edu for 1,2,4,6 or 12 hours at 28°C. Larvae were fixed overnight in 4% PFA, either immediately post-EdU labelling (progenitor analysis) or after 3–14 days (radial cohort analysis). For the radial cohort analysis, we restricted our analysis to those cell cohorts where the participating sections were fully intact, and cells could be counted across all three layers. To identify cell nuclei and the central border of the CMZ for cell counting, 2.5 µg/mL Hoechst was applied to retinal sections for 45 minutes at room temperature prior to mounting slides in Aquapolymount. EdU positive cells in the CMZ were counted in and averaged for 3–4 central retinal sections/larva.

**Sema3fa overexpression.** For Sema3f overexpression, we engineered a *sema3fa* heat shock construct, *hsp70:sema3fa:p3E-MTpA* in *pdestTol2CG2* with *myl7:egfp* as the transgenesis marker [70]. One cell-stage zebrafish embryos were injected with a solution consisting of 12.5-25 ng/µl plasmid and 25 ng/µl transposase mRNA. Larvae (5 dpf) with eGFP positive heart expression were left for 60 minutes at 38°C and then moved back to 28°C. Two days later the larvae were bathed in 500 µM Edu for 8 hours at 28°C and then fixed overnight in 4% PFA. Retinal cryostat sections were immunolabelled for Myc (Sema3fa-Myc) and underwent a Click-iT reaction to identify Sema3fa-myc- and EdU- expressing cells, respectively. Numbers of EdU-positive cells (3–4 sections/larva) were counted in dorsal CMZ that showed no (Myc-) or considerable Myc (Myc+) immunolabelling. Alternatively, heat shock was applied at 52 hpf and 24 hours later embryos were fixed and retinal sections immunolabelled for Myc and Cdh2 and analysed as described below.

**Cdh2 analysis.** Cryostat sections (12 µm) of 72 hpf WT and *sema3fa*<sup>ca304</sup> retinas, or the heat-shocked Sema3fa overexpression larvae, were immunostained with a polyclonal antibody against Cdh2 (N-cadherin, Abcam, 1:250) [71]. Images were captured of 2–4 central retinal sections/embryo. The central CMZ edge was defined by the presence of the inner and outer plexiform layers. The area of the bright Cdh2 immunopositive label was represented as a percentage of the area of the CMZ, and the percentages for several sections averaged for each individual embryo.

**Imaging.** Images of WM labelling were collected on a Stemi SV 11 stereoscope using an AxioCam HRc camera with ZEN (Zeiss) software. Retinal sections were imaged on a Zeiss Axioplan 2 microscope with an AxioCam MRc camera using ZEN (Zeiss) software. Images of FISH in WM and on retinal sections were taken using an LSM 900 confocal microscope with an AxioCam 305 camera using ZenBlue (Zeiss) software. Images were processed for brightness and contrast using Adobe Photoshop 2020.

## Statistics

The underlying numerical data for all graphs and summary statistics in the text have been deposited in a Dryad repository: https://doi.org/10.5061/dryad.pc866t21r [72]. All statistical analyses were completed on Prism 10 software (GraphPad). An unpaired, non-parametric Mann Whitney U test was used for statistical analyses of data, except where indicated. A Levene's test for the equality variance was used to compare the standard deviations of average cohort distributions between retinal laminae for individual eyes. Data sets for quantitation were analyzed, where possible, in a fashion where the researcher was masked to the genotype/experimental condition. N refers to the number of repeated experiments for a data set. The number of biological replicates or individual larvae used for a data set is the "n".

## Supporting information

**S1 Fig. Expression of mRNA for Sema3fa and potential receptors in the CMZ. A-B)** Double FISH on a cryostat retinal section for *ccnd1* (A) and *sema3fa* (B) mRNA in the CMZ (outlined with blue dashed line), with the *ccnd1* domain represented in B by a dotted yellow line. *sema3fa* mRNA largely overlaps with *ccnd1*, but neither gene is expressed in the most distal CMZ where the retinal stem cells reside (white arrows). **C-J)** Plastic sections of whole mount 72 hpf (C, E, G, I) and 7 dpf (D, F, H, J) larvae processed for in situ hybridization with antisense riboprobes for *nrp1* (C-F) and *plxna* (G-J) receptors.
(TIF)

## Acknowledgments

The authors would like to thank Dr. Sarah Childs for the use of her fish facility.

## Author contributions

**Conceptualization:** Amira Kalifa, Rami Halabi, Sarah McFarlane.

**Data curation:** Amira Kalifa, Carrie L. Hehr, Sarah McFarlane.

**Formal analysis:** Amira Kalifa.

**Funding acquisition:** Sarah McFarlane.

**Investigation:** Amira Kalifa, Carrie L. Hehr, Katelyn L. Shewchuk, Risa Mori-Kreiner, Shaelene Standing, Rami Halabi, Sarah McFarlane.

**Methodology:** Amira Kalifa, Carrie L. Hehr, Rami Halabi, Sarah McFarlane.

**Project administration:** Amira Kalifa, Sarah McFarlane.

**Resources:** Sarah McFarlane.

**Supervision:** Sarah McFarlane.

**Visualization:** Katelyn L. Shewchuk, Shaelene Standing.

**Writing – original draft:** Amira Kalifa.

**Writing – review & editing:** Carrie L. Hehr, Katelyn L. Shewchuk, Risa Mori-Kreiner, Shaelene Standing, Sarah McFarlane.

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
