## [Decision Letter · Decision Letter 0]

Response to Reviewers Revised Manuscript with Track Changes Manuscript

Aimée Dudley

Editor-in-Chief

PLOS Genetics

Anne Goriely

Editor-in-Chief

PLOS Genetics

**Additional Editor Comments (if provided):**
**Journal Requirements:**
**Reviewers' comments:**

**Comments to the Authors:**

Reviewer #1: In this manuscript, Kalifa and colleagues investigate the role of Semaphorin3fa (Sema3fa) on retinal progenitor proliferation and differentiation in the zebrafish eye. They find that a null mutation in sema3fa results in disorganization of the retinal stem cell niche – the ciliary marginal zone (CMZ). Progenitors in the CMZ express sema3fa as well as known receptors. The CMZ has been extensively characterized for gene expression and the positioning of progenitors at different preneurogenic stages. The CMZ can be divided into 4 zones based on proliferative capacity of the progenitors and marked by gene expression. The authors use in situ mRNA hybridization analysis of the mutants to show disorganization of the CMZ – specifically marker expression for Zones 3 and 4. The number and density and polarity of progenitors, however is unaffected. However, cell cycle markers and the neurogenic potential (as assessed by EdU pulse labeled clonal composition analysis) is altered. Last the authors show that markers of cell adhesion within the niche are upregulated. Overall, the conclusion that sema3fa, a pleiotropic secreted factor, organizes gene expression within the CMZ and promotes appropriate adhesive properties that facilitate normal retinal neurogenesis. This is more of a ‘marker in the sand’ type study that lays the groundwork for future more mechanistic experimentation. In general the data shown are of high quality and conclusions are in line with results. However, there are concerns with sample sizes from some studies (particularly in key experiments in which changes within the mutant are rather subtle. I also feel the negative results from the gain-of-function studies of semaphorin3fa are hard to interpret and do not add much. These and other critiques and suggestions are listed below.

1. Sample sizes of N=1 or 2 is no acceptable for studies in Figures 2, 4, 7.

2. The heat shock, over-expression studies are difficult to interpret as no effect was found. Perhaps mosaicism or extent of expression was too low. A positive control or better – germ-line transgenic analysis might be needed. In general, I did not find the studies compelling in design and could simply be removed.

3. With regard to much of the negative data (or differences in nasal vs temporal), maternal expression could be a confound. Were maternal-zygotic embryos/larvae ever examined. This seems very doable given the non-lethal nature of the mutation.

4. In Figure 6 – it seems that while the mutant clones are smaller, they are more numerous in number (15 clones from 14 wild-type retina ; 21 clones from 9 mutant retina). This is worth discussing.

5. In Figure 6A-B and C-D , are these different sections of the same sample/clone? If not, it looks like the clone shown in Figure 6C is from a rod progenitor, which was said to be excluded from the analysis.

6. In interpreting the smaller clone sizes in Figure 6, the authors might consider the fact that RGC, the first neuron type generated, was over-represented. Perhaps this shift in neurogenic competence ‘depleted’ progenitors resulting in smaller clone sizes. Related, it is worth discussing whether the disorganization in the niche altered the probabilities of symmetric vs asymmetric type divisions within clones. This is another mechanism that determines overall clone size and composition.

7. The term “masked analysis’ instead of ‘blinded’ is the preferred.

8. Lots of expression (and some other) data listed as ‘data-not-shown’. This should be included as supplemental figures.

Reviewer #2: The manuscript by Kalifa et al. identifies sema3fa as a novel regulator of CMZ neurogenesis in the postembryonic zebrafish retina. Interestingly, the authors demonstrate that in sema3fa mutants the typical lineage progression of stem cells and progenitors is disorganized and CMZ cells have compromised cell identities and disruptions in cell proliferation, even though the size and density of the mutant CMZ appears normal. The authors provide additional data showing that the contribution of new retinal cells from the mutant CMZ is impacted over time leading to small eyes in juveniles. Furthermore, it is proposed that enhanced cell adhesion among mutant CMZ cells is the main defect that causes the disorganized positioning of progenitor cells within the niche and their defective neurogenic behavior in early larval stages of retinal development.

Studies on CMZ neurogenesis over many years have yielded very good descriptions of patterning and cell fate progression in this stem cell niche. However, insight into specific mechanisms affecting the behavior of progenitor cells within the niche have been more limited. The present study offers a potentially valuable contribution to improving our mechanistic understanding of CMZ neurogenesis. For example, implicating sema3fa-dependent signalling in CMZ cell repulsion and/or adhesion is novel and significant. Nonetheless, there are several shortcomings related to: the rationale of certain experiments; the specific analyses used; and the need for additional complementary evidence to further support the main conclusions.

Major comments:

(1) The conclusions regarding changes in CMZ progenitor cell cycle are confusing and a clearer interpretation is required. In Fig 2 the authors report a reduction in PCNA ‘area’ in the mutant CMZ. In Fig 3, the authors report that the size and density of the mutant CMZ is normal throughout early larval stages. Together, these data would suggest that there are fewer cycling cells within the CMZ population. However, an increase in the number of EdU+ cells in the mutant CMZ is reported in Fig 2, and an increase in the ‘area’ of vsx2 and ccnd1 expression as a proportion of the CMZ is reported in Fig 3 and Fig 4, which would indicate an increased fraction of cells that are actively proliferating. This seems contradictory. The authors speculate in the Discussion that these data might suggest that mutant cells are stuck in S phase or G2, but this seems incongruent with the PCNA expression data (reduced) and the normal numbers of mitotic (pHH3+) cells indicating that mutant CMZ cells progress through to mitosis normally. Perhaps PCNA and EdU double labelling (or EdU cumulative labelling) could be used to more precisely discern changes in CMZ cell cycle kinetics in the mutants.

(2) The data on zonal expression patterns within the CMZ is interesting and it does appear that the proportion of stem cells might be expanded in the mutant relative to progenitors. It was also interesting that CMZ cells in mutants had altered identities with misexpression of genes that normally convey a cell fate progression toward differentiation. However, the main challenge with these interpretations is the reliance on in situ expression ‘area’. Using an area measurement as a proxy for the number of cells with a specific cell fate might be problematic in this context given the variation in the in situ signal, which is non-quantitative. For example, the authors show that in situ area changes in col15a1b and atoh7 positively correlate with RT-qPCR data, but in contrast the ccnd1 and cdkn1c data negatively correlate. While it is true that expression levels per cell might be different than the spatial distribution of expressing cells, the data presented do not clearly distinguish between these possibilities occurring in the mutant CMZ compared to controls. The increased overlap in the in situ expression domains of zone 2 with zone 3 genes in the mutants (there is also overlap in WT) might suggest that the markers of cell fate progression are altered but it’s not clear to me how this reflects both the “phase of neurogenesis” and “location of cells” within the CMZ, as the authors concluded (lines 303-305).

(3) Imaging data representing the over-expression of sema3fa-myc in Fig 2 (Fig 2R-T) is of relatively poor quality and it is difficult to discern the degree to which the myc+ transgene is expressed in EdU+ progenitor cells. It looks like most of the EdU+ cells do not express myc, perhaps due to the variegated heat-induced expression, and that might be reflected in the data showing no change in proliferation. I’m not sure this is an unambiguous test of whether transient sema3fa expression can induce changes in CMZ cell proliferation.

(4) Eye size is reported to be normal in the sema3fa mutants at 72hpf as well as at 10dpf, yet significant changes to CMZ progenitor cell proliferation is reported starting as early as 72hpf. The lack of evidence for changes in eye size during the first month would suggest that overall eye size might be uncoupled from changes in CMZ neurogenesis throughout the larval period in this mutant and is contrary to the authors’ main conclusion that defects in sem3fa-dependent CMZ neurogenesis is a cause of eye growth defects. Perhaps changes in eye size in mutant juvenile fish are a result of other growth processes in the eye, such as choroid and ciliary/iris growth, tissue stretching, etc., in addition to a reduction in CMZ neurogenesis. As it stands, the evidence for this correlation is not convincing. If it is a contributing factor, then perhaps showing eye size changes between 10dpf and 1mpf would help to emphasize the slow, protracted nature of this correlation making it more plausible, especially if changes are observed just after 10dpf.

(5) The pulse chase experiments reported in Fig 7 indicate that in the mutant, fewer new cells are added to the retina from the CMZ. The authors test the hypothesis that sema3fa is a positive regulator of repulsion, or negative regulator of cell adhesion, within the CMZ leading to mutant cells being less motile. To do this, they first induce the over-expression of sema3fa and look for proximity of transgene (myc+) expressing cells with neighboring cells expressing sema3fa receptor Plxna1. The rationale for this experiment is unclear. Plxna1 receptors appear to be expressed in all the CMZ cells (Fig 1L, O), so it is not surprising that there are no differences in proximity. Is there an increase in the level of plxna1 receptor expression rather than an increase in area of expression? Can the authors validate this with RT-qPCR or other methods? Also, the control used in this experiment is constitutive GFP expression rather than a heat-inducible construct like the sema3fa-myc. It would be more appropriate to use a heat inducible control.

(6) Perhaps a better test of a potential sema3fa repulsion mechanism for CMZ cells would be to induce the over-expression and then perform an EdU pulse-chase experiment to see if cohorts of labeled cells show increased motility as they emerge from the CMZ. The alternative explanation suggested by the authors was that mutant CMZ cells have increased adhesion. If so, then the same sema3fa over-expression experiment should result in a decrease in expression levels of cell adhesion markers. Was this examined?

(7) The authors note that there are no changes in cell death in the embryonic sema3fa mutant retina. However, it is possible that one reason for the reduced size of the EdU+ cohorts in the pulse-chase experiments is due to increased cell death of newborn cells in the larval mutant retina. The authors should check for this possibility using an apoptotic marker.

(8) The data represented in the graphs in Fig 6J-M is confusing. Is standard deviation being shown on the y-axes? The data are analyzed by MW statistics which accounts for ranks between the data points. Are the authors performing a test of significance on the standard deviation of the data? This is not clearly explained, but if I understand the goal of this analysis correctly, I recommend that the authors use a non-parametric ANOVA to analyze this data for the type of inferences they are trying to make.

Minor comments:

(1) The expression of sema3fa at 1mpf appears different using the DNP probe (Fig 1S) vs the DIG probe (Fig 1V). This might just be the choice of representative images, but the sema3fa-DIG expression more closely overlaps with postmitotic/differentiating progenitors rather than the actively proliferating progenitors. Can the authors comment?

(2) The Crb2a and aPKC expression data in Fig 4 exemplify the polarized nature of the of the CMZ neuroepithelial cells and are not specific stem cell markers as suggested by the authors. These apical markers are expressed throughout the CMZ and ONL in embryonic and early larval retina.

(3) For the EdU pulse-chase experiments described in Fig 6 and Fig 7, the authors refer to “clones” of EdU+ cells. Strictly speaking, the analysis is not a clonal analysis of lineages derived from single-labeled CMZ cells. Instead, it is a “cohort” of progenitors that are initially labeled with the EdU pulse, and these cells (and their progeny) are followed over the next two weeks. I recommend the authors clarify this description for accuracy. Also, in Fig 6A-D, the image panels are referred to as “14 day pulse” but it should read “14 day chase”.

Reviewer #3: PGENETICS-D-24-01000

Kalifa et al.

In this paper, the authors investigate which extracellular factors may be involved in the spatial patterning of progenitors in the ciliary margin zone (CMZ) of the zebrafish retina. While it has been known for many years that the CMZ contains different domains that contain specific subtypes of cells that contribute to the continuous growth of the retina throughout the fish life (stem cells, amplifying progenitors, committed precursors), little is known about the mechanisms governing the patterning of the CMZ. In this nice study, the authors show that Semaphorin3fa (Sema3fa) is required for the spatial patterning of the CMZ. They show that CMZ progenitors express mRNAs for various Sema3 receptors (e.g. nrp2a, nrp2b and plxna1) and loss of sema3fa does not affect the number of CMZ progenitors or the size of the CMZ, but the spatial organization is disrupted, which leads to disrupted generation of retinal cell types in the appropriate proportions and numbers. Consequently, Sema3f mutants display reduced eye size in juvenile fish. Using Edu labeling and clonal analysis, the authors go on to show that the dispersion of CMZ progenitors and their progeny is reduced in the absence of Sema3fa, and they show changes in expression of adhesion molecules. Based on these findings, the authors propose a model in which Sema3fa secreted by CMZ progenitors reduces adhesive interactions, which allows for the lateral progression of progenitors through the niche and proper generation of new retinal cells throughout the fish life.

Overall, this is carefully designed and executed study presenting interesting results that generally support the conclusions. The paper makes a significant advance in our understanding of spatial patterning of the retinal CMZ, an important stem cell niche involved in the regulation of adult tissue homeostasis and growth. Even though this CMZ niche does not exist in the adult mammalian eye, recent work has shown that a CMZ exist in the developing mouse (PMID: 28011038; PMID: 28009286), and even human retina (PMID: 38670973), and contributes to neurogenesis. Combined with the finding that Sema3fa is expressed in the developing rat retina, as stated by the authors, this suggest that Sema3fa may be involved in the spatial patterning of the embryonic mammalian CMZ as well. More generally, this work makes conceptual advance that further our understanding of spatial patterning of stem cell niche that could be extrapolated to other regions of the CNS, and even outside. It will be interesting to determine whether Sema3fa is also involved in patterning other niche regions. Below, I suggest a few point that the authors should consider, as it may help further improve the paper.

1) The authors use in situ hybridization to study expression of Sem3fa and its receptors in the CMZ at 72hpf and 7dpf in Figure 1. While the general expression pattern does appear to support the authors’ conclusion that Sema3fa is expressed in CMZ progenitors, while the receptors are expressed in cells closer to differentiated central retina, co-labelling with CMZ markers would help make this point more solid. Even though cyclinD1 was used to identify proliferating cells, markers more specific for zones 1, 2 and 3 would be useful here. For example, the authors suggest that lack of labelling for Sema3fa in the most distal part of the CMZ suggest that stem cells do not express Sema3fa, but positive labeling of this domain would be more convincing to suppor this conclusion. Also, there are many mentions of “data not shown” in this section, which is now unusual given that space is not an issue. I would suggest showing this data.

2) The authors may want to consider presenting the data on eye size (Figure 2) after showing disruption of spatial patterning in the CMZ of Sema3fa mutants (Figure 3). I feel like it would be a nicer flow to show what happens in the CMZ in absence of Sema3fa before showing the resulting outcome of this disorganization. Just a suggestion.

3) The authors studied the impact of Sema3fa inactivation on the regulation of eye size, which is an important function of the CMZ. But another important function of this stem cell niche is the regenerative response after injury. I was wondering if this is also affected in Sema3fa. If not, it would suggest that different mechanisms are at play, which would be quite interesting. If the authors have data on this or can perform the experiment easily, it seems to me like an important question to address. If not, then discussion of this point may be sufficient.

4) In Figure 2 and 7, the authors use a myc-tagged version of Sema3fa to determine whether it acts directly on progenitors to stimulate proliferation (Fig. 2) or as a repulsive cue (Fig. 7). In both experiments, the authors fail to report any changes by the expression of the Sem3fa-myc, arguing against a direct role in proliferation or repulsion. However, these are negative results and a positive control to show that the Sema3fa-myc is functional and can elicit an expected response is missing. This is important to add given that the results obtained are negative with this tool.

**Have all data underlying the figures and results presented in the manuscript been provided?**

Reviewer #1: **No: ** Excel sheets of quantitative data should be provided per journal standards.

Reviewer #2: **No: ** Numerical data underlying graphs and summary statistics was not provided in a spreadsheet as supporting information.

Reviewer #3: Yes

PLOS authors have the option to publish the peer review history of their article (what does this mean? ). If published, this will include your full peer review and any attached files.

**Do you want your identity to be public for this peer review?** For information about this choice, including consent withdrawal, please see our Privacy Policy .

Reviewer #1: No

Reviewer #2: No

Reviewer #3: No

**Figure resubmission:****Reproducibility:** To enhance the reproducibility of your results, we recommend that authors of applicable studies deposit laboratory protocols in protocols.io, where a protocol can be assigned its own identifier (DOI) such that it can be cited independently in the future. Additionally, PLOS ONE offers an option to publish peer-reviewed clinical study protocols. Read more information on sharing protocols at https://plos.org/protocols?utm_medium=editorial-email&utm_source=authorletters&utm_campaign=protocols

---

## [Decision Letter · Decision Letter 1]

Dear Sarah,

We are pleased to inform you that your manuscript entitled "Semaphorin 3f and post-embryonic regulation of retinal progenitors" has been editorially accepted for publication in PLOS Genetics. Congratulations!

Yours sincerely,

Cecilia

Cecilia Moens

Academic Editor

PLOS Genetics

Pablo Wappner

Section Editor

PLOS Genetics

Aimée Dudley

Editor-in-Chief

PLOS Genetics

Anne Goriely

Editor-in-Chief

PLOS Genetics

Comments from the reviewers (if applicable):

Reviewer's Responses to Questions

**Comments to the Authors:**

Reviewer #1: Authors have thoughtfully addressed all of my concerns. I have no further critiques.

Reviewer #2: Overall, I think the authors did a very good job of addressing reviewer comments. This has led to additional data supporting their conclusions and clearer interpretations of some of the original data.

Reviewer #3: The authors have addressed my questions and I have no more comments on the manuscript. Congrats on an interesting study!

**Have all data underlying the figures and results presented in the manuscript been provided?**

Reviewer #1: **No: ** I believe PlosGen requires the tabulated data files for all the quantitative experiments.

Reviewer #2: Yes

Reviewer #3: Yes

PLOS authors have the option to publish the peer review history of their article (what does this mean? ). If published, this will include your full peer review and any attached files.

**Do you want your identity to be public for this peer review?** For information about this choice, including consent withdrawal, please see our Privacy Policy .

Reviewer #1: No

Reviewer #2: No

Reviewer #3: No

**Data Deposition**

http://datadryad.org/submit?journalID=pgenetics&manu=PGENETICS-D-24-01000R1

**Press Queries**

---

## [Editor Report · Acceptance letter]

PGENETICS-D-24-01000R1

Semaphorin 3f and post-embryonic regulation of retinal progenitors

Dear Dr McFarlane,

We are pleased to inform you that your manuscript entitled "Semaphorin 3f and post-embryonic regulation of retinal progenitors" has been formally accepted for publication in PLOS Genetics! Your manuscript is now with our production department and you will be notified of the publication date in due course.

With kind regards,

Judit Kozma

PLOS Genetics

On behalf of:
